



# Drivers of seasonal and event scale DOC dynamics at the outlet of mountainous peatlands revealed by high frequency monitoring

Thomas Rosset [1], Stéphane Binet [2], Jean-Marc Antoine [3], Emilie Lerigoleur [3], François Rigal [4], Laure Gandois [1]

[1] EcoLab, Université de Toulouse, CNRS, Toulouse, 31326, France
[2] Université d'Orléans, CNRS/INSU, BRGM, ISTO Orléans, 45071, France
[3] GEODE, Université de Toulouse, CNRS, Toulouse, 31058, France
[4] IPREM, Université de Pau et des Pays de l'Adour, CNRS, Pau, 64000, France

Correspondence to: Thomas Rosset (thomas.rosset@ensat.fr)

**Abstract.** Peatlands store about 20 % of the global soil organic carbon stock and are an important source of dissolved organic carbon (DOC) for inland waters. Recent improvements for *in situ* optical monitoring revealed that the DOC concentration in streams draining peatlands is highly variable, showing seasonal variation and short and intense DOC concentration peak periods. This study aimed to determine the variables driving stream DOC concentration variations at seasonal and event scales. Two mountainous peatlands (one fen and one bog) were monitored in the French Pyrenees to capture their outlet DOC concentration variability at a high frequency rate (30 min). Abiotic variables including precipitation, stream temperature and water level, water table depth and peat water temperature were also monitored at high frequency and used as potential predictors to explain DOC concentration variability. Results show that at both sites, DOC concentration time series can be decomposed into a seasonal baseline interrupted by many short and intense peaks of higher concentrations. The DOC concentration baseline is driven, at the seasonal scale, by peat water temperature. At the event scale, DOC concentration increases are mostly driven by water table increases within the peat at both sites. Univariate linear models between DOC concentration and peat water temperature or water table increases show greater efficiency at the fen site. Water recession times were derived from water level time series using master recession curve coefficients. They vary greatly between bog and fen but also within one peatland site. They partly explain the differences between DOC dynamics in the studied peatlands, including porewater DOC concentrations and the links between stream DOC concentration and water table rise. This highlights that peatland complexes are composed of a mosaic of heterogeneous peat units distinctively producing or transferring DOC to streams.

## 1. Introduction

Aquatic carbon transfer from terrestrial ecosystems to inland waters is receiving increasing attention as it plays a major role in the watershed carbon balance (Webb et al., 2018) and in the global carbon cycle (Cole et al., 2007; Drake et al., 2017). The origin of aquatic carbon has been tracked and wetlands have been shown to be the main organic carbon suppliers to rivers at both local (Hope et al., 1997; Laudon et al., 2004; Ledesma et al., 2017) and continental scales (Hope et al., 1994; Spencer et al., 2013). Peatlands are specific wetlands which have accumulated organic matter through slow vegetation decomposition



processes (Joosten and Clarke, 2002; Limpens et al., 2008). Peatlands grow under different climates (Broder et al., 2012; Dargie et al., 2017; Gorham, 1991; Page et al., 2011) and store about 21% of the total global soil carbon stock (Leifeld and Menichetti, 2018). Stream outlets of peatlands have been monitored at different latitudes (Billett et al., 2006; Leach et al., 2016; Moore et al., 2013) in order to quantify and understand the aquatic carbon transfer between these organic carbon rich

pools and their draining streams. Dissolved organic carbon (DOC) is a key component of these fluxes as it contributes to more than 80% of the aquatic carbon exported from peatlands (Dinsmore et al., 2010; Hope et al., 2001; Müller et al., 2015; Roulet et al., 2007). At the outlet of peatlands, DOC is not only considered for its role in the carbon balance but also because it may be an issue for water treatment quality (Ritson, 2015) and a conveyor of potentially harmful elements along inland waters (Broder and Biester, 2017; Rothwell et al., 2007; Tipping et al., 2003).

Variability in the DOC concentration signals at the outlet of peatlands has been observed at the inter-annual (Fenner and Freeman, 2011; Köhler et al., 2008), the seasonal (Leach et al., 2016; Tipping et al., 2010) and even the event scales (Austnes et al., 2010; Dyson et al., 2011). Different drivers have been identified depending on the latitude of the studied peatland sites and the time scale considered. DOC concentrations were found to be negatively correlated with discharge in boreal systems (Köhler et al., 2008), positively correlated with discharge in temperate areas (Clark et al., 2007) or non-correlated with

discharge in mountainous areas (Rosset et al., 2019). Temperature was also reported as an important driver of seasonal variations of DOC concentration in field (Billett et al., 2006) and mesocosm (Pastor et al., 2003) experiments since DOC production is boosted by a greater microbial activity during warmer periods. Higher temperatures were also shown to enhance evapotranspiration from peatland resulting in a rise in DOC concentration in peat porewater and stream waters during dry summer periods (Fraser et al., 2001). Finally, other studies documented the importance of water table level fluctuations

(Bernard-Jannin et al., 2018; Kalbitz et al., 2002; Strack et al., 2008) and acrotelm oxygenation (Freeman et al., 2001) in DOC production and mobilization to streams.

DOC concentration monitoring at the outlet of peatlands has generally consisted in a weekly or monthly stream water sampling routine (Clark et al., 2008; Juutinen et al., 2013). Higher frequency sampling has been restricted to specific high precipitation events (Austnes, 2010; Clark et al., 2007) or snowmelt (Laudon et al., 2004). Recently, new optical *in situ* sensors (Rode et

al., 2016) were used to track DOC concentration at a high frequency rate (~30 minutes) at the outlet of peatlands (Koehler et al., 2009; Ryder et al., 2014; Tunaley et al., 2016), highlighting the strong variability of the DOC concentration signal over a year. While diel DOC concentration cycles have been analyzed under steady hydrological conditions (Tunaley et al., 2018), no analysis has yet been performed to understand the high frequency variability of the DOC concentration at a multi-year scale. Mountains host many small peatland areas that are often neglected in global peatland assessments but which drastically

influence stream chemistry in headwater catchments (Rosset et al., 2019).The harsh mountainous climatic conditions (from the montane to the alpine belt (Holdridge, 1967)) and the relief of those areas generate high gradients of different abiotic parameters (temperature, precipitation, hydrology). Furthermore, seasonal climatic conditions are contrasted, making it possible to differentiate seasonal from event scale stream DOC concentration variability. The present study aimed at disentangling the mechanisms driving DOC concentration observed at the outlet of two peatlands in the French Pyrenees





mountains. A bog and a fen were monitored for stream DOC concentration using an optical high frequency *in situ* sensor placed at their outlet. The scientific objectives of this study were (1) to identify the main abiotic parameters driving stream DOC concentration variability at each site, (2) to identify the temporal scale of these drivers, and (3) to compare the DOC concentration patterns between a bog and a fen.

## 2. Study sites

The peatland of Bernadouze (Fig.1-b) is situated in the Eastern part of the French Pyrenean mountains (42°48'9" N; 1°25'25" E). The peatland lies at 1343 m.a.s.l. It belongs to a 1.4 km² watershed on limestone rocks dominated by the Mont Ceint =2088 m.a.s.l. and particularly steep (average slope=50%). From a post-glacial lake, a fen developed for 10 000 years at Bernadouze site, reaching a peat accumulation depth of 2 m in average and more than 9.5 m at extreme locations (Jalut et al., 1982; Reille,

1990). As surficial runoff contributes to the water supply of the peatland, it is considered as a soligenous (minerotrophic) fen (Joosten and Clarke, 2002). The fen is subject to an oceanic climatic influence but weather conditions can locally be contrasted due to the specific mountainous topography. For the years 2015 to 2018, the mean annual temperature was 7.9±0.3 °C and the mean annual precipitation was 1797±265 mm. Sub-zero temperatures and snow events are regularly observed at Bernadouze site from mid-October to mid-May with a snow cover lasting around 85 days (Gascoin et al., 2015) from December to April

and sometimes exceeding 2m in height. Beech forest is the dominant vegetation cover in the watershed, except for the highest grassland areas (> 1800 m) and the 4.7 ha of the peatland. Vegetation on the peatland is mainly composed of species characteristic of minerotrophic peatlands such as *Carex demissa* and *Equisetum fluviatile*. However, some ombrotrophic species such as *Sphagnum palustre* and *Sphagnum capillifolium* are observed on the southern part of the peatland, forming small hummocks and revealing a progressive disconnection with the stream and the water table supply. Logging activities were

carried out during the autumn 2016 in the lowest forested area surrounding the peatland.

The peatland of Ech (Fig.1-c) culminates at 710 m.a.s.l. in the west-central part of the French Pyrenees (43°4'59" N; 0°5'39" W). Dominated to the North by mount Cossaout (1099 m.a.s.l.), the peatland depends on a 0.86 km² watershed principally composed of grasslands and grazing areas. The bog area is 5.3 ha and the peat deposit reaches 3.3 m in the center (Millet et al., 2012). Peat formation started about 8200 years ago from a post glacial lake dammed by a recessional moraine in the South

(Rius et al., 2012). The peatland is classified as a bog since the surface vegetation depends only on water supplied by precipitation. The site experiences a mountainous oceanic climate characterized by an average annual temperature of 11±0.2 °C and an annual precipitation of 1242±386 mm (data from 2015 to 2018). Sub-zero daily mean temperatures are rare (~10 days a year) and snow events are sparse in Ech. From the model of (Gascoin et al., 2015), the average duration of snow cover does not exceed 10 days at this altitude in the Pyrenees. The vegetation observed is typical of ombrotrophic bogs with a large

blanket of *Sphagnum Capillifolium* and *Sphagnum Compactum*. Small birches and hummocks of *Molinia caerulea* have started to develop within the peatland. Many burning events have been reported on the peatland since its formation (Rius et al., 2012). Nowadays, agro-pastoral practices still use fire to limit the vegetation height and *Molinia caerulea* extension. The last burning



event at the Ech site occurred 8 weeks before the stream monitoring in April 2017 and concerned the North Eastern half of the peatland. A second burning event occurred in February 2019 in the Western area of the site. It was decided to stop data acquisition just before the fire to avoid potential shifts in DOC concentration induced by this anthropogenic disturbance (Brown et al., 2015).

## 3. Material and methods

### 3.1. Site instrumentation

This article presents high frequency data monitored from the 1st September 2015 to 31st December 2018 at Bernadouze site and from 22nd May 2017 to 19th February 2019 at Ech site. Precipitation (liquid and solid) and air temperature were recorded every 30 minutes at Bernadouze (Gascoin and Fanise, 2018) and every 60 minutes at Ech by automatic weather stations located respectively 300 and 15 meters from the peatlands in open areas. At both sites, sensor failures prevented data acquisition and gap-filling models were used to complete the datasets. For precipitation data, a linear model ($r^2$=0.99, p-value< 0.01) based on cumulative precipitation recorded in Saint Girons (414 m.a.s.l, 42°58'58"N, 1°8'45"E) was built to generate total daily precipitation in Bernadouze. A similar model was built in Ech ($r^2$=0.99, p-value< 0.01) based on data recorded in Ossen (517m, 43°4'0"N, 0°4'0"W). Missing air temperature data were estimated at Bernadouze from a linear regression model ($r^2$=0.99, p-value< 0.01) based on data monitored at the same rate under the forest canopy 100 m away from the main weather station. In Ech, daily mean temperatures were estimated using a linear regression model ($r^2$=0.88, p-value< 0.01) with daily mean temperature recorded in Tarbes (360 m.a.s.l. 43°10'55"N, 0°0'2"W).

At the outlet of each peatland, a multiparameter probe (Ysi Exo2, USA) measured fluorescence of the organic matter (fDOM, λexcitation=365±5 nm / λemission=480±40 nm), turbidity, water level and temperature every 30 minutes. Wiper sensors prevented the optical sensors from biofouling before each measurement and the probes were inspected and calibrated monthly. At both sites, a network of piezometer wells (8 in Bernadouze and 4 in Ech) were used to record hourly the water table depth and the water temperature with automatic probes (Orpheus Mini Water Level Logger, OTT HydroMet, Germany). Piezometer locations were defined to be representative of the different topographic and vegetation surfaces observed on each peatland (hummocks, lawns, river banks).

### 3.2. Water sampling and DOC calibration

Grab water sampling was performed every two weeks at the outlet of Bernadouze peatland and every two months at the outlet of Ech. Piezometer wells ([1.5, 2.5] m depth (Fig.1)), were used to sample peat water on four occasions (2013, 2014, 2015, 2018) in Bernadouze and on two occasions (2017, 2019) in Ech during stream baseflow periods. Grab water was collected using a manual peristaltic pump and was directly filtered on site using 0.22 µm cellulose acetate filters (GSWP04700, Merck-Millipore, USA). To avoid contamination from cellulose, the first millimeters of filtered water were discarded. Water samples were brought back to the laboratory in a cool box and were stored at 6°C until analysis. High resolution water sampling was



performed during 9 flood events at the outlet of Bernadouze and once at Ech using automatic water samplers (ISCO 3700, USA) to collect water during various hydrological conditions. Each flood sampling event consisted in collecting 24 samples of raw water (950 mL) at a frequency defined thanks to the observed timelag of discharge (1 hour for rainfall and 4 hours for snowmelt driven flood events). Flood water samples were collected within the 48 hours following the previous sampling and

processed as grab water samples at the laboratory.

For all samples (grab and flood samples), non-purgeable organic carbon (NPOC, referred to hereafter as DOC) concentration was analyzed in filtered samples after acidification to pH 2 with a TOC-5000A analyzer (Shimadzu, Japan). The quantification limit was 1 mg. L$^{-1}$. Reference material included ION-915 and ION96.4 (Environment and Climate Change Canada, Canada). The fluorescence of DOM (fDOM) data was explored for potential adjustments for temperature, inner filter effect and turbidity

(Downing et al., 2012; de Oliveira et al., 2018; Watras et al., 2011). fDOM data were corrected for temperature as described by (de Oliveira et al., 2018). The inner filter effect was adjusted at Ech for data showing absorbance values at 254 nm higher than 0.6 (de Oliveira et al., 2018). Lastly, fDOM data recorded during the turbidity events (>20 FNU) were ignored in the analysis as the fluorescence can be drastically attenuated by the presence of particles (Downing et al., 2012). High frequency DOC concentrations were calculated at each site using a site specific linear model ([DOC]=a*fDOM+b) linking corrected

fDOM data to DOC concentration in flood and grab-water samples. The two models are respectively described by the following parameters: (a=0.192, b=-0.031, number of observations =174, r²=0.93, p-value<0.001) for Bernadouze and (a=0.294, b=-1.39, number of observations =27, r²=0.78, p-value<0.001) for Ech.

### 3.3.  Water level fluctuation characterization

In order to provide an overall characterization of the peatlands, a mean water table depth, as well as a mean water temperature

was calculated at each site by averaging water table depths and water temperature data at a given time from the set of piezometer probes. Calculations were performed only when all sensors were running (94% of the time period in Bernadouze and 100% in Ech). Hereafter, the mean water temperature in the piezometers is assimilated to peat water temperature.

Master Recession Curve (MRC) analyses were performed on water table and stream level time series, using the MRCTools v3.1 software (Posavec et al., 2017). The MRC represents the average recession of the water level observed when only

discharge flow occurs (no recharge). An exponential master recession curve was used to adjust the observed average MRC and to define a specific recession coefficient (α, unit=day$^{-1}$) characteristic of each monitoring point (Eq.(1)).

$$Master\ Recession\ Curve \Leftrightarrow Water\ level = f(t) = K * e^{-\alpha t} \tag{1}$$

The exponential recession coefficient corresponds to the inverse of the average water recession time, called recession time, in the area of a piezometer or in a stream after a precipitation event. In the following, the recession time coefficient (1/α) is used

to characterize the hydraulic properties of peatlands and stream.



### 3.4. DOC peak selection and characterization

Peak selections in the DOC concentration time line were performed running Python 3.6 (Python Software Foundation, 2019) scripts using the function find_peak available in the SciPY Signal library (Jones et al., 2001) and the arithmetic mean of the DOC concentration signal (DOC_mean) as an input parameter. Peak selection criteria were: to reach DOC_mean concentration

and have a prominence higher than 0.25 times DOC_mean. Peaks occurring during an interval shorter than 12 hours apart were grouped under the highest DOC concentration peak. Each DOC concentration peak was defined by the time period delimited by the two nearest low points surrounding the peak event. Low points were located on the DOC concentration time lines by applying the find_peak function on the negatively transformed (-1*) DOC concentration signal previously processed with a Savitzky-Golay filter (window-length=23 and polyorder=2). Low points occurring during an interval shorter than 12 hours

apart were grouped under the lowest DOC concentration point. Lastly, the DOC peak period could be manually adjusted to fit or correct a peculiar peak pattern. A DOC concentration peak period was characterized by different metrics (Fig.2): its initial value corresponding to the DOC concentration of the low point at the start of the peak period (DOC_initial), its maximum value corresponding to the DOC peak value (DOC_max), its range (DOC_increase) which was calculated by subtracting the initial value from the maximum value and finally by the rising time duration (rising_limb) which separates the initial low point

time from the peaking time. In this study, initial values and increases of DOC were the targeted variables to be explained. Initial values of DOC were used to determine a DOC concentration baseline (Fig.2). The following classification was used to describe seasonal variations: winter (December, January, February), spring (March, April, May), summer (June, July, August), and autumn (September, October, November).

### 3.5. Explanatory variables selection and characterization

In order to investigate DOC concentration variabilities (at two temporal scales: peak event and seasonal), nine explanatory variables were selected. These variables were extracted from DOC concentration, stream water level, mean water table, stream temperature, air temperature, peat water temperature and precipitation time lines measured at each site. DOC peak events were reported on each time line and variables were calculated for each event using metrics similar to those previously described in the DOC peak characterization section (Fig.2).

The variables were chosen because they have been reported in the literature to have an explanatory potential for stream DOC concentration variability (Table 1). Two categories of variables were distinguished depending on whether the process they described was related to the production of DOC within peatlands or to the transfer of DOC from peatlands to streams. After sensitivity tests and in accordance with the observations of (Tunaley et al., 2018), a mean of seven days prior to the event was defined as the best operator to characterize air and stream water temperatures.





### 3.6. Correlation and statistical modeling

Relationships between targeted variables (DOC_increase and DOC_initial) and the explanatory variables were investigated using ordinary least squares (OLS) multiple regression analyses. Prior to the analyses, variables which did not satisfy a normal distribution were log or square root transformed to improve normality (Table 1). Multicollinearity was assessed among all the predictors using Pearson correlation with a threshold |r<0.7| following (Dormann et al., 2013). When two variables were found to be collinear, we selected the one that displayed the highest absolute correlation with the targeted variables. Then at both sites, all variables were standardized to a mean of zero and a standard deviation of one to derive comparable estimates in the following analysis. We performed a backward stepwise selection procedure on the full model (i.e. the model including the variables retained after removing multicollinearity) to capture the best set of variables explaining each targeted variable. At each step of the procedure, the non-significant variables (p-value>0.05) with the highest p-value were dropped from the model and the resulting reduced model was re-evaluated. This process was continued until there were no non-significant variables remaining in the final model. To account for the time dependency of the variables in the analyses, time was also included as an explanatory variable in the full model. This variable corresponds to the duration which separates each DOC peak event from the start of the time line. Residuals of the final models were surveyed in order to detect deviations from normality and homoscedasticity and to identify outliers. No specific deviations or outliers were detected. Model residuals were also checked for autocorrelation to verify the absence of any cyclical variation in the variables set. When more than one variable was retained in the final model , the relative contribution of each variable was assessed using hierarchical variance partitioning (Chevan and Sutherland, 1991). According to the previous predictor selections for the MLR models of DOC concentration increases (DOC_increase), OLS regression analyses were performed at each piezometer plot of a peatland site, replacing the mean water table increase variable by the specific water level increase values of each plot. Similar OLS regression analyses were performed at the outlet of streams by replacing the mean water table increase variable by the stream water level increase when necessary. R² and relative importance (%) of the water level increase variable were reported for each OLS regression tested. All the analyses were undertaken in R (R Core team, 2019) using the package *rms* (Harell, 2019) and *relaimpo* (Groemping and Matthias, 2018).

### 4. Results

#### 4.1. Description of temperature, precipitation, water table and DOC time series

Climatic variables are contrasted between the two studied areas. In 2018, temperatures were higher in Ech than in Bernadouze with an annual mean air temperature, water temperature and peat water temperature respectively of 11.3, 10.7,11.9 °C compared to 7.9, 7.1, 7.7 °C. This contrasted with total precipitation which reached 2151 mm in Bernadouze and 1140 mm in Ech. In these steep mountainous headwaters, short and intense flood events were triggered by strong precipitation events and/or the snowmelt. Over the whole timelines, the maximum and mean of the stream water level were respectively 1.36 and 0.35 in





Ech and 0.81 and 0.10 m in Bernadouze. These short flood events were followed by recession sequences revealed by the slow decreases in the water table at both sites, especially in late summer and autumn (Fig.3-c). The average and minimum of the water table depth in the two piezometer networks were respectively -0.23 and -0.43 m at Ech, and -0.15 and -0.45 m in Bernadouze. At both sites, no clear relationship was observed between the stream and the water table time series. For a given

precipitation amount, the water levels responded differently depending on the season. For instance, a strong flood observed in the stream can be contiguous with a low or high water table rise (i.e. July 2016 and February 2017 events in Bernadouze) (Fig.3 b -c).

DOC concentration was highly variable at both sites during the monitored periods as highlighted by the numerous short DOC peak events (~30 hours duration) in the two time series (Fig.3 and Table 2). At Bernadouze site, DOC concentration peaks

showing higher values were more frequent from April to November while this was less obvious at Ech site where DOC concentration also peaked during winter. In 2018, the arithmetic means and flow weighted averages of DOC concentration were clearly higher at the outlet of Ech, reaching 7.1±6.1 and 4.6 mg L$^{-1}$, than in Bernadouze where they were 2.0±1.5 and 1.7 mg L$^{-1}$.

### 4.2. DOC concentration peaks characterization

Peak characterization (Table 2) revealed that the increases and maxima of DOC concentration peaks were on average two times higher in Ech than in Bernadouze. However, the ratio between the mean increase and the mean initial value of DOC concentration was higher in Bernadouze (2.3) compared to Ech (1.9). DOC concentration peaks occurred more often at Bernadouze compared to Ech (0.24 vs 0.16 peak per day in average) while their duration was slightly longer (32±14 vs 28±16 hours). Rising limbs of DOC concentration peaks lasted on average 10±5 and 13±14 hours at Bernadouze and Ech respectively

and they were slightly longer than the stream water rising limb averages monitored at the outlet of the two peatlands. In contrast, rising limb duration of the water table in Ech was clearly longer (22 ±12) compared to Bernadouze (13±7 hours).

General mean and seasonal means of initial DOC concentrations were 2.5 and 3.1 times higher at Ech compared to Bernadouze (Table 3). However, at both sites, DOC_initial showed a clear seasonal variability. The lowest values were observed in spring

and the highest in autumn while in summer and winter DOC concentration was close to the annual mean. DOC peak event frequencies also varied at the seasonal scale (Table 3). The highest frequencies were reported in autumn at both sites. The lowest peak frequencies were observed in winter at Bernadouze and in summer at Ech.

### 4.3. DOC concentration variations models

Prior to multiple regression analyses, the air temperature over 7 days, the stream water level maximum and the initial level of

the water table were excluded from the analysis because of their strong correlation with other variables (Pearson's correlation $|r > 0.7|$) (Fig.A1). Multiple linear regressions (MLR) followed by backward stepwise selections showed that respectively 55% and 44% of the seasonal variation of DOC (DOC_initial) was explained by the final models at Bernadouze and Ech (Table 3).



Peat water temperature was reported as an important predictor at both sites (72% of the variance explained by the final model at Ech and 44% at Bernadouze). In Bernadouze, variance is similarly explained by the time between two peaks (44%). Along the two years of monitoring in Ech, the strong DOC concentration values observed during the dry autumn 2018 (Fig.2) created a positive general trend in the DOC concentration baseline. This peculiar trend drastically influenced the statistical analysis

and consequently the variable "time" became a significant predictor at the seasonal scale. Considering the high relative importance of the peat water temperature in the two final models, two simple linear models (Fig.4 a) were built based on this variable to illustrate the seasonal DOC concentration behavior in Bernadouze (slope=0.08, intercept=-0.16, n=231, R²=0.26, p-value<0.001) and in Ech (slope=0.10 intercept=0.50, n=100, R²=0.27, p-value<0.001). For the DOC concentration increase final models, water table increase was the most important variable at Bernadouze (67% of the variance explained) and the

single variable at Ech. In Bernadouze, other variables such as water temperature, water level increase and the time between two peaks were significant enough to be integrated in the reduced DOC_increase model. The $R^2$ associated to the models varied strongly between the two sites, reaching 0.77 in Bernadouze and only 0.27 in Ech. Since water table increase was the main explanatory variable for the DOC concentration increase model, two simple linear models were built (Fig.4b) with the following parameters in Bernadouze (slope=8.44, intercept=-1.06, n=231, R²=0.68, p-value<0.001) and in Ech (slope=6.39,

intercept=0.84, n=100, R²=0.27, p-value<0.001).

### 4.4. Relationships between DOC dynamics and recession time

In the fen of Bernadouze the recession times in the peat ranged from 15 to 77 days whereas in the bog of Ech they were longer, ranging from 53 to 143 days (Fig.5). Stream recession times were clearly shorter at both sites reaching 4 days in Bernadouze

and 9 days in Ech. Results of the OLS regressions conducted at each water level monitoring plot using DOC increase final models, revealed that recession time influenced the model's efficiency (Fig.5 a). Piezometers characterized by shorter recession times showed greater determination coefficients R² (Fig.5 a). Water table increase was the most important predictor (pie charts Fig.5 a) for all piezometer plots, contributing at least 47% of the explained variance of the DOC increase models. In Bernadouze, the model based on stream level was weaker (R²=0.37) than the models based on water table data while in Ech

the model based on stream level was unable to explain at all the DOC increase variation (R²=0). Recession times showed a positive relationship with DOC concentration measured in the piezometer and in the streams. Stronger concentrations were observed for longer recession times (Fig. 5 b).

### 5. Discussion

### 5.1. Methodological breakthrough

To our knowledge, this is the first time that stream DOC concentration and abiotic drivers, including water table depth fluctuations, have been analyzed at peatland sites on a multi-year period at such a frequency (30 min). Previously, DOC





concentration variability had already been investigated but this was done either at lower frequencies (Clark, 2005; Dawson et al., 2011) or during shorter periods (Austnes et al., 2010; Koehler et al., 2009; Tunaley et al., 2016; Worrall et al., 2002). The originality of this study was to sequence DOC concentration peaks, considered as biogeochemical "hot moments " (McClain et al., 2003) within the peatland carbon cycle, to disentangle event and seasonal drivers of DOC concentration variability.

Thanks to the high frequency monitoring, a large number of events (252 peaks in Bernadouze and 101 peaks in Ech) were captured at all seasons (Table 2), enhancing the representativeness of both seasonal and event scale statistical models.

### 5.2. Peat water temperature controls seasonal DOC concentration baseline

Clear seasonal variations in the DOC concentration baseline were observed at both sites (Fig.3 and Table 2). These variations are mostly driven by peat water temperature (Table 3). Baseline DOC concentration increased in late spring, peaked in autumn,

decreased during winter to reach the lowest levels in early spring. Similar seasonal DOC concentration patterns have been observed at other peatland sites in temperate regions (Austnes, 2010; Broder and Biester, 2015; Clark et al., 2005; Tunaley et al., 2016; Worrall et al., 2006; Zheng et al., 2018) or after the snowmelt event in boreal areas (Jager et al., 2009; Köhler et al., 2008; Laudon et al., 2004; Olefeldt and Roulet, 2012; Whitfield et al., 2010). The higher DOC concentration observed in summer could be explained by evapotranspiration processes which concentrate solutes in stream water. However, the

evapotranspiration rates in these mountainous environments are low (<300 mm year$^{-1}$) compared to precipitation (>1200 mm year$^{-1}$) and should not drastically influence the seasonal DOC concentration baseline. In Bernadouze, DOC concentration remained extremely low when the fen was snow-covered and it did not drop drastically during the spring snowmelt as has been observed in boreal areas (Laudon et al., 2004; Leach et al., 2016). This pattern can be explained by (1) the low initial DOC concentration which prevents a clear dilution being observed during the snowmelt event, (2) the snowmelt regime in this

Pyrenean catchment which may be less sudden than in boreal regions and occurs from the early snow deposit to the beginning of the growing season, continuously diluting the low winter DOC production within the peatland.

Temperature is often identified as a DOC concentration driver in peatlands at the seasonal scale (Billett et al., 2006; Clark et al., 2008; Dawson et al., 2011; Koehler et al., 2009). Warmer temperatures directly enhance DOC production by stimulating vegetation and microbial activity (Kalbitz et al., 2000; Pastor et al., 2003). Warmer temperatures are also indirectly linked to

DOC production processes in temperate and northern peatlands since they often correspond to dry periods that lower water table levels. When the water table decreases, the "enzymic latch" (Freeman et al., 2001) is initiated on a greater volume of oxygenated peat and enhances DOC production within the upper peat layers. In these two mountainous peatlands, peat water temperature was shown to be the main predictor of the seasonal stream DOC concentration baseline (Table 3). Similar DOC concentration relationships with peat water temperature have already been described in an acidic fen in France (Leroy et al.,

2017) and in blanket peatlands from the North Pennine uplands in the UK (Clark et al., 2005); however, in these cases DOC concentrations were measured in peat porewater. A complementary study in the North Pennines (Clark et al., 2008) showed that peat porewater DOC concentrations and stream DOC concentration were strongly correlated, meaning that, by extension, the relationship between peat temperature and stream DOC concentration could be verified for these sites.





In the present study, both stream and peat pore water DOC concentrations were higher in the bog than in the fen (Table 2 and Fig.5 b). This is consistent with mesocosm (Pastor et al., 2003) or field (Chanton et al., 2008; Chasar et al., 2000; Moore, 1988) porewater observations. Differences in the seasonal DOC concentration baseline (DOC initial) models can be explained by the variation in the leachable DOC pool within peatlands. The influence of temperature on DOC production appears similar

at both sites since the slopes of the univariate models were comparable (Fig.4 a). However, the lower altitude and the southern orientation of Ech peatland influenced overall temperatures positively compared to Bernadouze. Therefore, DOC production was able to occur at a high rate over longer periods at Ech. DOC production also depends on vegetation types (Armstrong et al., 2012; Leroy et al., 2017; Vestgarden et al., 2010). Sphagnum species, which are dominant on bogs, produce relatively less labile and reactive DOC than vascular plants, which are more abundant on fens (Chanton et al., 2008; D'Andrilli et al., 2010).

In Bernadouze, contrary to the initial hypothesis (Table 1), time between peaks was a negative significant predictor in both seasonal and event DOC concentration models (Table 3). This is considered as an indirect consequence of the seasonal temperature control on DOC concentration. Indeed, snow cover and the low temperatures associated to high water table positions prevent the occurrence of DOC peaks in winter, creating large time gaps between two events (Table 2) of low initial values. In contrast, DOC production is amplified in warmer periods resulting in more frequent stream DOC concentration

peaks starting at higher initial values. In Ech, where average annual temperatures are higher, the initial hypothesis was verified since DOC concentrations were stronger in autumn after the long summer times between peaks (Table 2). However, the variable was not significant enough to be integrated in any final model.

### 5.3. Water table increase controls DOC concentration peaks at the event scale

This study, coupling high frequency stream DOC concentration and water table depth monitoring at both peatland sites,

revealed that water table increase is a strong predictor of stream DOC concentration increase at the event scale (Table 3 and Fig. 4 b). Stream DOC concentration variability at the event scale has been investigated in terms of discharge but never in terms of water table variation. Several studies have reported stream DOC concentration increases at the outlet of peatlands during flood events (Austnes, 2010; Ryder et al., 2014; Tranvik and Jansson, 2002; Yang et al., 2015), whereas others showed dilution during high flow events (Clark et al., 2007; Grayson and Holden, 2012; Laudon et al., 2004; Worrall et al., 2002). In

Bernadouze, stream water level only poorly contributed to explaining the variability of DOC increases during flood events (Table 3 and Fig.5 a) and in Ech, it did not contribute at all. This is in line with the studies reporting a non-linear flow-DOC concentration relationship at the outlet of peatlands (Roulet et al., 2007; Tunaley et al., 2016).

Water table is usually considered as a DOC production driver as it controls the oxygenated acrotelm volume (Billett et al., 2006; Freeman et al., 2001; Ritson et al., 2017). Therefore, different studies attempted to quantify the effect of water table

position on DOC production rate in peatlands. On the one hand, Pastor et al., (2003) observed no DOC concentration variation in the stream water after a water table decrease in a fen and a bog mesocosm. On the other hand, increasing DOC concentrations were observed during the re-wetting phase of the acrotelm at fen sites in Germany (Kalbitz et al., 2002), in Canada (Strack et al., 2008) and in the USA (Hribljan et al., 2014). Finally, Clark et al., (2009) reported similar observations after re-wetting



peat cores in controlled laboratory conditions. Our results are in line with these studies, since they highlight the role of rising water table for DOC transfer from peatlands to streams.

Nevertheless, this study contrasts with former studies which considered single time water table position or seasonal variations at peatland sites. At the seasonal scale, water table level is only an indicator of the potential occurrence of a DOC peak event.
For instance, DOC peak events are prevented or minimized during winter and early spring since the water table is high (Fig.2 b), whereas DOC concentration increases are strongest during the low water table periods in summer and autumn. Using high frequency monitoring, it was possible to quantify the rise of water in the upper peat layers and the volume of peat rewetted during an event. The models developed in this study show that stream DOC concentration is proportional to the volume of rewetted peat (Fig. 4 b). This means that the water table increase within the peatland is the limiting factor of DOC concentration
increase in the streams. This is in line with practices for degraded peatland restoration, where a general rise of the water table is recommended to limit water table increases and induced DOC concentration peaks at their outlets (Höll et al., 2009; Strack and Zuback, 2013).

### 5.4. Contrasted DOC dynamics related to recession times

Spatial analysis of water table variation within the peatland revealed that the studied sites are composed by several peat units,
characterized by contrasted recession times. In these mountainous peatlands, recession times are related to DOC dynamics, driving model efficiency between DOC concentration increase and water table rise and explaining DOC concentration in peat pore water.

In average the bog of Ech presented a longer recession time (111 days) than the fen of Bernadouze (20 days). However, the range of recession times is wide in a peatland complex presenting simultaneously bog and fen characteristics. For instance, a
specific unit in the fen of Bernadouze was characterized by a long recession time of 77 days. This unit showed surface bog vegetation and topographic patterns but was surrounded by typical fen units characterized by clear lower recession times (Fig.5). Thus a peatland complex must be considered as a patchwork of different units and not as a uniform peat entity.

At the event scale, the univariate model between DOC concentration and water table increase showed a non-negligible intercept at Ech contrasting with the model of Bernadouze (Fig.4 b). Therefore, in Ech, DOC concentration increases could occur
without being associated with water table increases. In this case, DOC is transferred from the upper peat layers via fast runoff flows without any water table level fluctuation. Such a phenomenon is consistent with the hydraulic properties associated to the bog and estimated via the recession times (Fig.5). Indeed, the long recession times of bog units suggest slow percolation rates. During precipitation events, the infiltration capacities of the surface peat layers of bog units are rapidly saturated, enhancing surface runoffs and preventing or delaying water table increases (Table 2). In contrast, DOC stored in the upper
peat layers of fen units is transferred to the stream by fast percolating water, which raises the water table levels and supplies the sub-surface flows in these zones (Fig.6). This explains why the DOC increase final model based on water table increase is particularly efficient for fen units characterized by short recession times (Fig.5 a). Recession times also explain the differences in peat porewater DOC concentration observed between bog and fen sites. In the fen, recession times are short, meaning that





the upper peat layers are rapidly washed by precipitations, inducing sudden DOC pool depletions of the peat porewater (Fig.3 c). At the bog site, DOC stored in the upper peat layers is slowly released to the stream after precipitation events and contributes to maintaining a high stream baseline (Fig.3 c) and peat pore water DOC concentrations (Fig.5 a).

Thus, stream DOC concentration modelling at the outlet of peatlands must account for different proportions of fen-like or bog-like units in peatland complexes to fit the real seasonal and event DOC concentration variability. Each unit supplies DOC to the stream at a different rate depending on its volume, distance from the stream and recession time (Fig.6). This end member mix concurs with the model of Binet et al., (2013) describing event and seasonal water table variability in peatlands using a double porosity parametrization. In that sense, recession time appears as a new physical parameter able to characterize peatland units otherwise than by the binary typology: bog or fen. This would surely improve the efficiency of hydrological and biogeochemical models. In the case of peatland complexes characterized by long recession times, further investigations of peatland runoffs and sub-surface flows are needed, analyzing denser and stream directed piezometer transects in order to build stronger DOC concentration models.

## 6. Conclusion

This study reports an analysis of the stream DOC concentration variability at the outlet of two mountainous peatlands. Multi-year *in situ* high frequency (30 minutes) monitoring revealed that at both sites, DOC concentration time series can be decomposed in a seasonal baseline interrupted by many short, intense peaks of higher concentrations. At the seasonal scale, DOC concentration baseline variations are mainly explained by peat water temperature which controls integrative DOC production processes within the peatland. During the "hot moments" of peak events, DOC concentrations are well explained at both sites by water table increases within the peatlands.

Recession time is a relevant parameter to explain peat porewater DOC concentration and the different model performances observed between bog and fen sites. Recession time assessments in different locations on the two studied sites showed that peatlands are composed of different units presenting contrasted water recession properties. Thus, peatlands should not be considered as uniform landscapes. Distinct peatland units within the same peatland complex contribute differently to the DOC transfer processes to inland waters. Recession time assessment in piezometers appears to be a simple and promising tool to investigate hydrological processes occurring in peatlands over time and space. Indeed, water table time series are often under-used and only account for a seasonal mean or minimum depth. Assessing recession times on peatlands is a first step to taking peatland water table dynamics into consideration and to explaining potentially related biogeochemical processes.





## 7. Data availability

The data used in this manuscript are described and available on the Pangaea® data repository at:

https://doi.pangaea.de/10.1594/PANGAEA.905838

## 8. Author contribution

| CASRAI role | Rosset | Binet | Antoine | Lerrigoleur | Rigal | Gandois |
|---|---|---|---|---|---|---|
| 1 *Conceptualization* | x | x | | | | x |
| 2 *Data curation* | x | | | x | x | |
| 3 *Formal analysis* | x | x | | | x | x |
| 4 *Funding acquisition* | | x | | | | x |
| 5 *Investigation* | x | x | x | x | | x |
| 6 *Methodology* | x | x | x | | | |
| 7 *Project administration* | x | x | | | | |
| 8 *Resources* | x | x | | x | | x |
| 9 *Software* | x | | | x | x | |
| 10 *Supervision* | | x | | x | | x |
| 11 *Validation* | x | x | | | | x |
| 12 *Visualization* | x | x | | x | | x |
| 13 *Writing – original draft* | x | x | | | | x |
| 14 *Writing – review & editing* | x | x | x | x | x | x |



## 9. Competing interest

The authors declare that they have no conflict of interest.

## 10. Acknowledgment

All the carbon analyses were performed at the PAPC platform (EcoLab). The authors wish to thank: L.Plagnet for permitting
the access to the peatland of Ech, V. Payré-Suc, F. Julien, D. Lambrigot, W. Amblas for assisting in stream organic carbon
concentration analysis; F. De Vleeschouwer, D. Allen, P. Durantez Jimenez, T. Camboulive for assisting in water sampling;
G.Susong and the Regional Natural reserve of Pibeste-Aoulhet for piezometer maintenance, S. Gascoin, P. Fanise, the CESBIO
laboratory and the OSR Toulouse for providing the meteorological data, D. Galop for assisting in site preparation and
communication with local policy makers, E. Rowley-Jolivet for English language assistance.

This project was made possible with the support of the LabEx DRIIHM , French programme "Investissements d'Avenir" (ANR-
11-LABX-0010) which is managed by the ANR and funds the Ph.D of T. Rosset; LabEx DRIIHM OHM Haut Vicdessos/
Haute Vallée des Gaves,  REPLIM OPCC,  ANR JCJC TRAM (ANR JCJC 15-CE01-008 TRAM), which funded the
investigations at both sites.

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

**Figure 1: a) Location map of Ech Bog (brown plot) and Bernadouze fen (green plot) in South Western Europe. Satellite views of the peatlands of Bernadouze b) and Ech c) and location of the site instrumentation. Map source: Esri, DigitalGlobe, Geoeye, Earthstar Geographics, CNES/Airbus DS, USDA, USGS, AEX, Getmapping, Aerogrid, IGN, IGP, swisstopo, and the GIS User Community.**



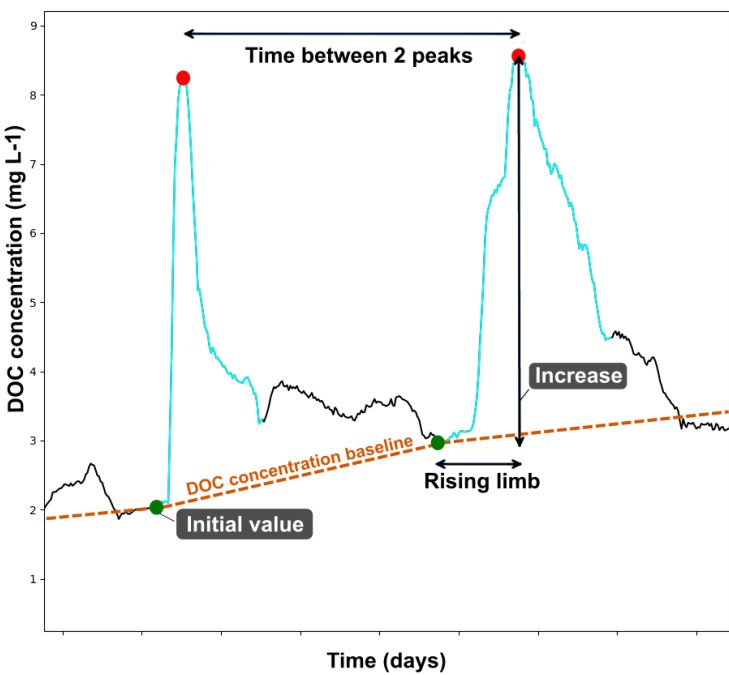

**Figure 2: Characterization of DOC concentration peaks. Peak events are identified on the DOC concentration time line in blue. Each DOC concentration peak event is defined by an initial concentration (green points) and a maximum one (red points). DOC concentration increase is calculated by subtracting the initial from the maximum concentration. The time between 2 maximum DOC**
5  **concentrations corresponds to the duration (seconds) separating two events and is used as an explanatory variable. The DOC concentration baseline (orange dotted line) corresponds to the time series defined by all the initial values of each DOC concentration peak.**



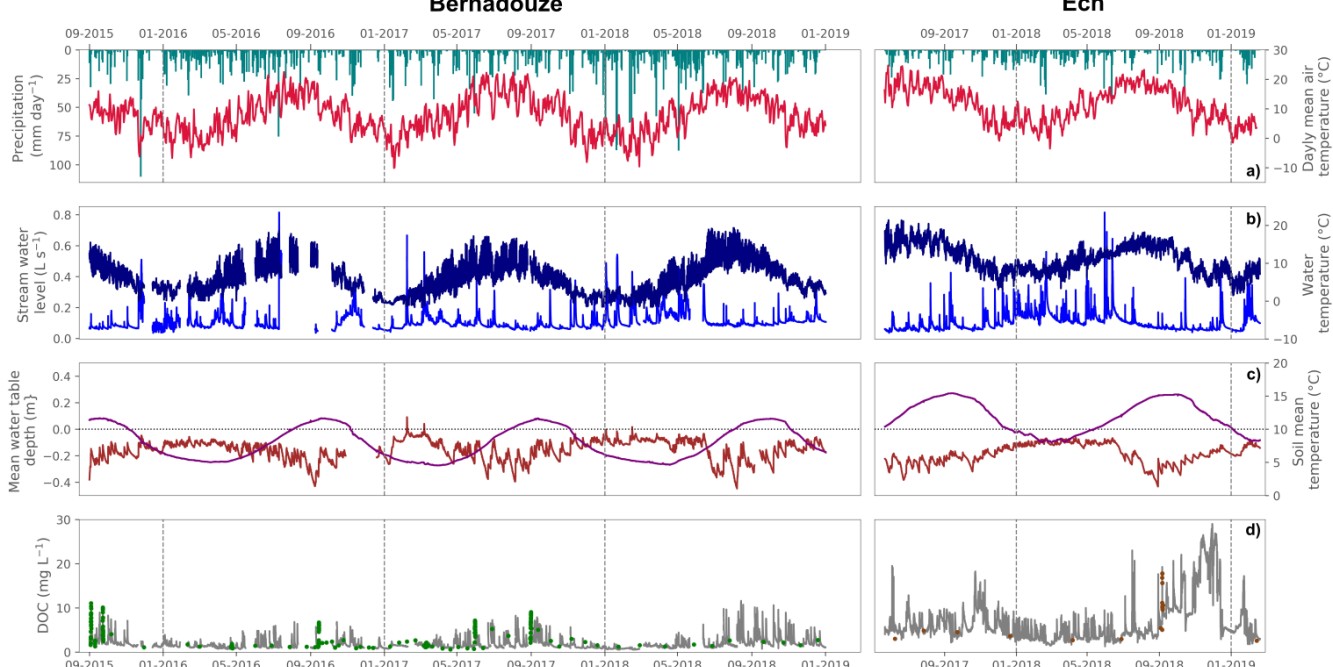

**Figure 3: Precipitation and air temperature (a), stream temperature and water level (b) high frequency DOC concentration (c), mean water table depth variation and peat water temperature (d). Time series observed at the outlet of the Bernadouze fen (left panel) from 1st September 2015 to 31st December 2018, at the outlet of Ech bog (right panel) from 22nd May 2015 to 13th February 2019. The vertical grey lines represent a change of year. Green (for Bernadouze) and brown (for Ech) plots in time series (c) refer to DOC concentration measured in grab water samples and automated flood samples.**



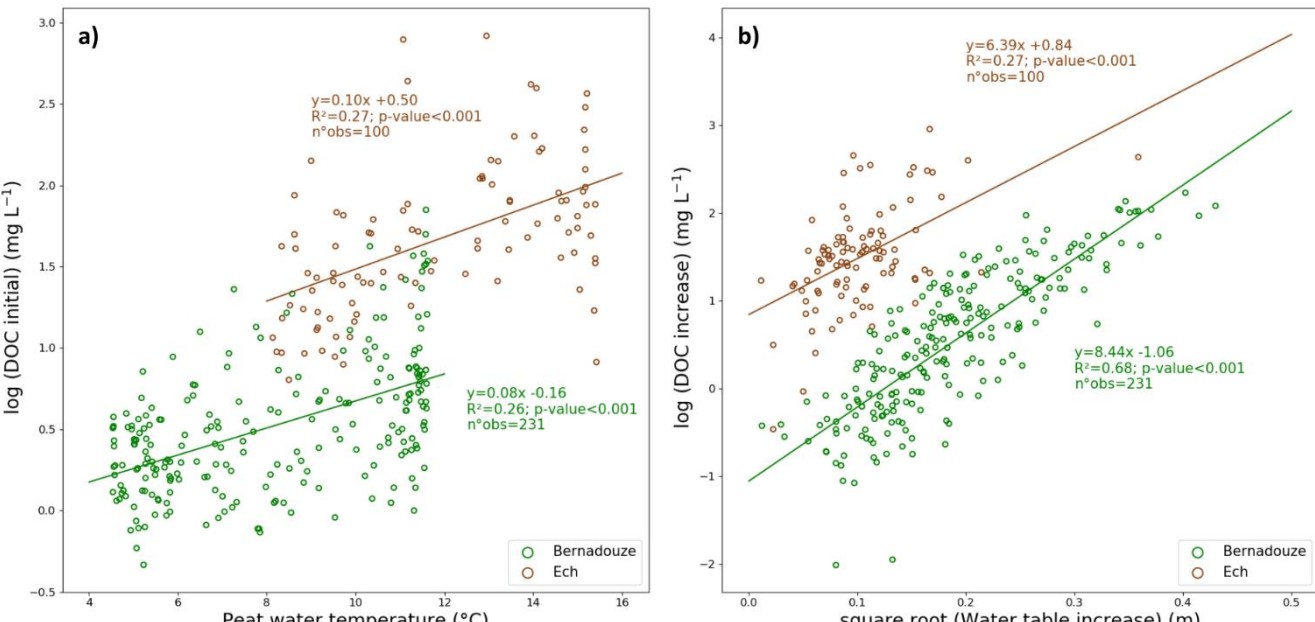

**Figure 4: Relationships between (a) peat water temperature and natural logarithm of DOC concentration initial value and (b) square root of water table increase and natural logarithm of DOC concentration increase during peak events at Bernadouze (green) and Ech (brown). Regression coefficients (intercept and slope), p-values and $R^2$ are given in each panel.**





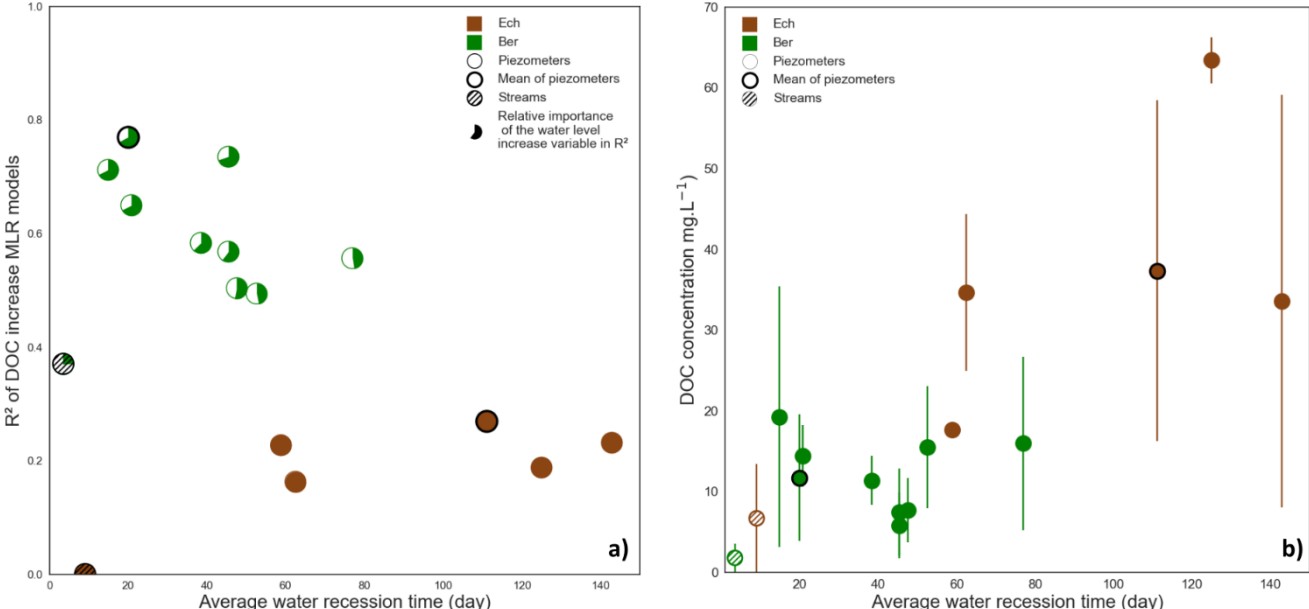

**Figure 5: Relationship between average water recession (WAR) time coefficients and a) the R² of the DOC_increase MLR models or b) the DOC concentration of each water level monitoring point at the peatland of Bernadouze (green) or Ech (brown). Piezometer plots correspond to solid circles and the mean of the piezometers at each site is surrounded in black, stream plots correspond to striped circles. Pie charts in a) represent the relative importance of the water level increase variable in the R² of each model. Markers in b) represent the mean measurements and vertical segments the standard deviations of DOC concentration at each location.**





**Figure 6: Schematic overview of a peatland complex. Size of the arrows corresponds to DOC quantity mobilized from distinct peatland units. The DOC concentration observed in the stream depends on the contribution of the different peat units within the peatland complex.**



**Table 1. Targeted and explanatory variables description**

| | Designation | Variable description | Hypothesis | References | Statistical transformation |
|---|---|---|---|---|---|
| | | | Targeted variables | | |
| Seasonal scale | DOC initial | DOC concentration at the start of a DOC peak event | Define the stream DOC concentration baseline | | Logarithmic |
| Event scale | DOC increase | Range between DOC concentration initial value and maximum observed during a DOC peak event | | | Logarithmic |
| | | | Explanatory variables | | |
| Production | Time between peaks | Duration between two DOC concentration peaks | Longer intervals between peaks promote DOC production and induce higher stream DOC concentration elevations during the next rewetting | (Fenner and Freeman, 2011; Ritson et al., 2017; Worrall et al., 2006) | Logarithmic |
| | Air temperature 7 | Weekly mean of the water temperature prior to the DOC peak event | High temperatures enhance microbial and vegetation activity which increase DOC production within the peat and DOC concentration in the stream | (Billett et al., 2006; Clark et al., 2005, 2008, 2009; Koehler et al., 2009; Pastor et al., 2003) | |
| | Water temperature 7 | Weekly mean of the water temperature prior to the DOC peak event | | | |
| | Peat water temperature | Water temperature observed at the beginning of the DOC peak event (from the mean water temperature of the piezometers) | | | |
| | Water table initial | Water table value at the beginning of the DOC peak event (from the mean water table level of the piezometers) | Initial water table value is an indicator of the non-saturated peat depth. A lower initial water table is related to a higher volume of oxygenated peat, where | (Bernard-Jannin et al., 2018; Billett et al., 2006; Clark et al., 2009; Fenner and Freeman, 2011; Ritson et al., | |



 

| | | | most of the DOC is produced. | 2017; Tunaley et al., 2016) | |
|---|---|---|---|---|---|
| Transfer | Stream level increase | Stream water level increase during the DOC concentration peak | DOC concentration increases with stream water elevations | (Austnes, 2010; Ryder et al., 2014) | Square-root |
| | Stream level maximum | Water level maximum during the DOC peak event | | | logarithmic |
| | Precipitation 1 | Total daily precipitation recorded during the rising period of the peak and the day prior to the DOC peak event | Precipitation triggers lateral transfer of DOC-rich water from peatland to surface water.<br><br>Amount of precipitation is assumed to be representative of the surface runoff | (Raymond et al., 2016) | Square-root |
| | Water table increase | Water table increase during the DOC peak event (from the mean water table level of the piezometers) | Water table rise promotes DOC transfer to the stream through sub-surface flows. The greater the re-wetted peat volume (water table range), the stronger the stream DOC concentration | (Clark et al., 2009; Kalbitz et al., 2002; Strack et al., 2008) | Square-root |





**Table 2. Time series and DOC concentration peak metrics in Bernadouze over the 1st September 2015 to 31st December 2018 period and in Ech over the 22nd May 2015 to 13th February 2019 period. Mean notations correspond to arithmetic means which are given with standard deviations.**

| | | Unit | Bernadouze | Ech |
|---|---|---|---|---|
| Time series | Days of study | Days | 1218 | 638 |
| | DOC data available | % time | 86 | 99 |
| | DOC (arithmetic mean) | mg L$^{-1}$ | 1.8±1.2 | 6.7±4.9 |
| | Discharge (arithmetic mean) | L s$^{-1}$ | 34.1±74.2 | 8.4±12.0 |
| | DOC concentration (flow weighted mean) | mg L$^{-1}$ | 1.6 | 5.0 |
| DOC concentration peaks | Number of peaks | | 252 | 101 |
| | DOC maximum (maximum) | mg L$^{-1}$ | 11.6 | 23.3 |
| | DOC maximum (mean) | | 4.3±2.2 | 11.1±4.6 |
| | DOC increase (maximum) | | 9.3 | 19.2 |
| | DOC increase (mean) | | 2.4±1.9 | 5.2±3.3 |
| | Water table increase (mean) | M | 0.04±0.03 | 0.01±0.01 |
| | DOC peak duration (mean) | H | 32±14 | 28 ±16 |
| | DOC peak rising duration (mean) | | 10±5 | 13±10 |
| | Stream water level rising duration (mean) | | 10±7 | 12±11 |
| | Water table rising duration (mean) | | 13±7 | 22±12 |
| DOC concentration baseline | DOC initial (mean) | mg L$^{-1}$ | 1,9±1.0 | 5,9±3.1 |
| | Autumn | | 2,5±1.2 | 7,9±3.4 |
| | Winter | | 1,7±0.7 | 5,3±3.5 |
| | Spring | | 1,4±0.4 | 3,5±1.1 |
| | Summer | | 1,7±0.9 | 5,6±1.2 |
| | Time between peaks (Mean) | H | 116±169 | 149±179 |
| | Autumn | | 97±144 | 133±132 |
| | Winter | | 196±221 | 140±219 |
| | Spring | | 122±214 | 152±212 |
| | Summer | | 105±111 | 180±172 |




**Table 3. Reduced models explaining DOC concentration during peak events (DOC_initial and DOC_increase) at the outlet of Bernadouze and Ech peatlands. Reduced models were obtained after a backward stepwise selection procedure applied on the full model (See details in Methods). Adjusted R² of each model are given as the predictors and their associated coefficient, p-value and R² contribution.**

| Response variable | Site | Adjusted R² of the reduced models | Reduced models | | | |
|---|---|---|---|---|---|---|
| | | | Coefficients | Predictors | p-value | R² contribution |
| DOC initial | Bernadouze | 0.55 | 0.62 | Peat water temperature | <0.001 | 0.24 |
| | | | -0.50 | Time between peaks | <0.001 | 0.24 |
| | | | 0.16 | Precipitation 1 | 0.002 | 0.02 |
| | | | -0.25 | Water temperature 7 | <0.001 | 0.03 |
| | | | -0.14 | Stream level increase | 0.009 | 0.02 |
| | Ech | 0.44 | 0.84 | Peat water temperature | <0.001 | 0.32 |
| | | | 0.24 | Time | 0.004 | 0.05 |
| | | | -0.33 | Water temperature 7 | 0.004 | 0.04 |
| | | | 0.16 | Precipitation 1 | 0.017 | 0.03 |
| DOC increase | Bernadouze | 0.77 | 0.74 | Water table increase | <0.001 | 0.52 |
| | | | 0.26 | Water temperature 7 | <0.001 | 0.17 |
| | | | 0.09 | Stream level increase | 0.019 | 0.07 |
| | | | -0.14 | Time between peaks | <0.001 | 0.02 |
| | Ech | 0.27 | 0.52 | Water table increase | <0.001 | 0.27 |



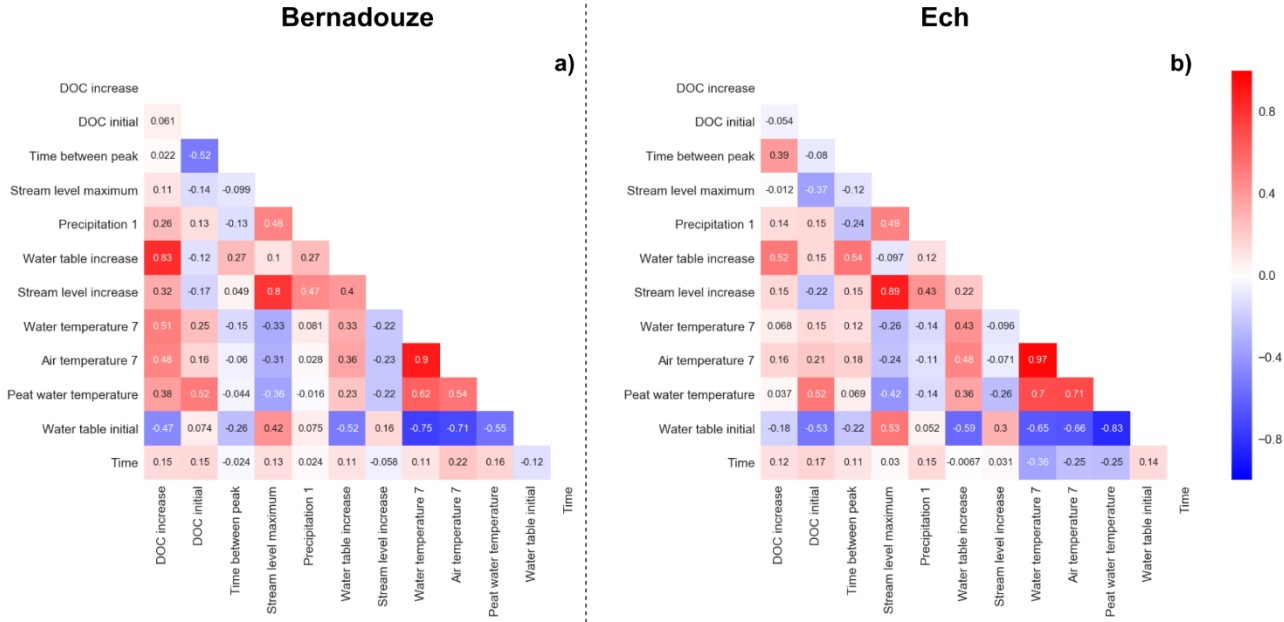

**Figure A1. Pearson correlation matrices between the DOC concentration targeted variables and common explanatory variables at Bernadouze a) and Ech b). In view of their strong correlation with other variables (Pearson's correlation |r > 0.7|), the air temperature over 7 days (air_temp_bf7d), the stream water level maximum (log_water_level_max) and the initial water table level (piezo_level_initial) were excluded from the analysis. The air temperature over 7 days was preferentially excluded compared to water temperature over 7 days because of data reliability (air temperature was gap-filled at Ech).**