# Peer review of "Drivers of seasonal and event scale DOC dynamics at the outlet of"

_Biogeosciences, 2019_

## Referee Comment (RC1) · Anonymous Referee #1 · 12 Nov 2019

General comments:

Rosset et al. reported high-resolution sensor data to investigate the mechanisms driving DOC concentration at the outlet of a bog and a fen in the French Pyrenees. The data and results are interesting. However, the paper can be improved further by explaining how complete are the sensor data, and providing discussion on how water temperature is related with the input and output of organic carbon in the bog and the fen. Specific comments are below, which the authors may consider when revising the manuscript.

Specific comments:

[Figure]

p. 4, line 10-24: What is the percent of data for which gap-filling models were used? Also, has there been any period of power outage? The merit of this paper is on the high-resolution 'sensor' data. Thus, the information is needed on the number (or the percentage) of data points that has been actually collected.

p. 5, line 8-: How accurate was the analysis? What was the recovery of the reference material?

p. 5, line 12-: If the data with >20 FNU were ignored, what is the percentage of those "ignored" data points compared to the total? Also, considering that [DOC] can be high with high flow, those data points are potentially important in interpreting the results. If included, could they change the conclusions? I think the graphs showing the relationship between the [DOC] and fDOM would be helpful. Can you add the graph as a supporting information?

p. 5, line 16: number of observations 174 vs. 27. Why are these so different?

p. 5, line 27: what is the K in the equation 1? Please explain the terms in the equation.

p. 6, line 11: Have you used "DOC_max" for the analysis? If not, why didn't you include it for the analysis?

p. 7, line 3-: So, did log- or square root-transformation satisfy the assumption? Was non-parametric analysis unnecessary?

Fig. 1: Is the boundary of the watershed for the 'outlets' correct? Watershed boundary can be delineated for any point of a stream using DEM data. The watershed area for the red circles should be larger than the boundary of the fen or the bog (orange lines in Fig. 1). I wonder the DOC dynamics at the outlets could be significantly influenced by non-wetland areas considering that the stream lines are extended beyond the orange lines.

Fig. 3: Is the purpose of this research on comparison between the fen and the bog? If so, which period should be used? The same overlapped period (May, 2017 to Jan.,

2019)? Or any period with available data? If you have chosen the second option (any period with available data) to maximize analysis power, why did you omit the period of Jan. 2015 to Sep. 2015 (Rosset et al., 2019, JGR-Bioigeosciences)?

Fig. 4: Interesting graphs. (a) When log (DOC initial) is ~2.0, the DOC initial should be ~ 100 mg/L. But, the maximum [DOC] in the Fig. 3d is ~30 mg/L. Why are these this so different? (b) What are the meanings of the y-intercept? When water table increase is 0, the log (DOC increase) is about -1 (fen) and +1 (bog). Then, DOC increase should be 0.1 mg/L (fen) and 10 mg/L (bog) even without the water table increase. What kind of mechanism is working?

Concentration of DOC shows the dynamic balance between the input and output of organic carbon. How water temperature is related with the input and output of organic carbon in the bog and the fen?

Fig. 5: The graphs include many information and are hard to digest. I recommend to leave essential information only and provide the rest as a supporting material. Or figure caption can include in-detail explanation on the symbols.

---

## Referee Comment (RC2) · Anonymous Referee #2 · 21 Nov 2019

The manuscript of Thomas Rosset et al. aims at disentangling drivers of DOC dynamics in different peatland catchments. It utilizes a spectrofluorometric probe to monitor high frequency DOC concentrations. Additionally, parameters like temperature, precipitation, stream and peat water level were recorded to explain DOC concentration variability by means of statistic modelling. The generated dataset is quite extensive and worth to be published in Biogeosciences. As I was really pleased by the title and abstract I must say that the main part of the manuscript is rather descriptive and lacks a clear message or novelty. A major drawback of this manuscript is the lack of discussion of major processes and drivers concerning DOC export from peatlands, like hydrological flow paths, biogeochemistry and hydrologic preconditions. In its current

state, the study is too much focused on the statistical results of their modelling, but no or little mechanistic explanation of the modelling results. What exactly happens during a rainfall/discharge/DOC concentration event in the different systems and concerning the hydrologic conditions? In general I think that this paper needs a stronger discussion on hydrologic flow paths in peatland systems. Flow paths are not discussed until the very end. I miss a description of e.g. a simple acrotelm/catotelm distinction, which provide different hydraulic conductivities and thus lead to a distinct hydrograph. I miss a discussion of pre-event conditions, or of hydraulic conductivities in general, contributing to different flow paths. The interpretation of the piezometer data is difficult, as there is no information provided about depth of installation, hydraulic conductivities and thus how to interpret recession times. I miss mentioning importance of pH (DOC solubility!), do you have data about this? It is mentioned that the fen site is on limestone, while bog systems are generally acidic systems. As the authors are throwing in the term "biogeochemical hotspot" in the end: I would be pleased to hear more about this earlier on in the manuscript. The introduction on P 2 second paragraph is rather superficial. Furthermore, I am very concerned by the representativeness of the bog site especially when it is compared to a fen as exemplary system (Scientific objective no 3, P3 line 3). There are several factors differing between the sites, besides just fen/bog: climate (e.g. 4 month snow covered – no snow hardly sub-zero temperatures), anthropogenic influence (burned – unburned). Additionally, mentioning agro-pastoral practices: does this mean the bog is used for grazing? Could these systems thus be considered as representative? Besides this, from the location maps I draw conclusions that apparently the monitoring spot also receives water which is not originating from the peatlands itself. Is there any data about it? Do you have any idea about the whole catchment and how much water contributes to the discharge that is not from the peatlands? This is one of my major concerns, as I feel like the authors completely disregard this. If the concentration pattern are driven by discharge from other areas, the discussion of concentration pattern and water levels at the monitoring spot and within the peatland would be difficult. Did you calculate also total DOC export fluxes next to the concentrations?

This would highlight, how important this carbon output is to the system, as the concentrations seem to be quite low. Besides from this: Did you also compare the Bernadouze and Ech sites over a time period, where data from both sites are available? This would help to access, if there is any bias by having two different time periods here. Weather conditions can be very different each year having a very dry or wet year or season, respectively. I am no native speaker myself, so it is not easy for me to criticize language issues. But even though in your acknowledgement you state a language assistance, I am sure that there are some unusual or wrong wordings in the text (e.g. confusing usage of "contrasted" "strong concentrations"). So, from my point of view this needs further editing. Furthermore, your expression is imprecise at some points. Please edit (e.g. header of Table 1; P5 Lines2-5: it is not clear what you mean; or speaking of "stream level increase") Please, check your figures for clear distinction and readability when not color printed (e.g. Fig 1 stream/peatland boundary, Fig. 5 Ech/Ber)

In summary I see a very valuable dataset of potentially high interest. However, in its current state the discussion seems too superficial and the study remains rather descriptive. Therefore, I recommend major revisions before this study can be accepted for publication.

Specific comments:

P1 L 10/P2 L2: A very common number I know is 30% of global carbon stock. Please check more references for the number you give.

P2 L 12-15: too simplistic statements. Drivers of DOC concentration are not dependent on latitude, but mostly on the system studied and climate

P 2 L 32: "seasonal climatic conditions are contrasted" what does this mean?

P3 L 19: What "logging activities" do you mean?

P4 L 21: In which depths are the piezometers connected to the peat body? This is very important if you talk about recession times and peat water temperature. Maybe

also interesting: What diameter have these wells? Or did you attempt to determine hydraulic conductivities by a slug test?

P5 L 2 "Flood sampling" is a weird expression. What about event sampling?

P5 L 12: when did "the turbidity events" occur? I assume this is mostly the case during high discharge. How much of your DOC event data is affected by this?

P5 L 11: Reference not correctly inserted in text, happens occasionally in the manuscript

P6 L 20: which variables were selected? Next sentences are unclear.

P7 L 31 "...1.36 and 0.35. in..." unit missing

P7 L 19: why? You did not introduce site heterogeneity before (Introduction). I would also like to have an introduction in hydrologic connectivity, so are the different spots relevant for the discharge? What about the rest of the catchment?

P8 L 4 ff: Discuss different pre-conditions, changes in hydrologic conductivity with depth

P8 L 9 ff: give conc. maxima. I am very surprised by the low mean values. When you have 2 and 7 mg/L mean DOC concentrations I am about to doubt the significance for the carbon balance. This only gives me the idea that you have a lot of water not originating from the peatlands itself. Calculating fluxes might be helpful here. You might check literature and compare

P8 L 31/32: Is this much? More than half cannot be explained by your model in Ech, other factors seem very important as well. Which could this be?

P 9 L 1: Peat water temperature dependent on depth?

P 9 L 9: speaking from "water table increase" and "water level increase" is confusing to me. Give more precise names, maybe include "stream" or "peat" for clear distinction

P9 L 19: This is a very simple statement and would be very odd, if this is not the case.

P9 L 27: what is a "strong concentration"?

P9 L 29: Please reword

P9 L 30: Is this important? What is the novel statement/finding of your study? So far I mainly see confirmation of former results

P 10 L 2 ff: Biogeochemistry? This is new and not mentioned before. Please introduce, the biogeochemical background or processes could be much more discussed in this paper

P 10 L 5: expression "Thanks to" sounds uncommon, please reword

P. 10 L 7 following: Chapter 5.2 is badly structured. If evapotranspiration is unlikely don't put it there so prominent. The discussion of this part is interrupted and starts again further below -> confusing

P11 L 1: Peat pore water DOC: You did not mention those in the result section. A short description/discussion about those concentrations would be nice.

P11 L 3 f: What differences? Explanatory variables? Leachable Pools is a good keyword, but please discuss this on your data and not switch directly to the next topic.

P11 L 8. This is indicative for the discussion. There is just an isolated statement that vegetation type plays a role for DOC production. How is this related to your findings?

P 11 L 21: This is not true! Please try some more literature search

P 11 L 22 ff: Discuss flowpaths. Try e.g. DOC concentration vs Q plots. You throw in "non-linear flow DOC concentration relationship", try to discuss this.

P 11 L 30: just single observations, discuss mechanisms

P 12 L 3-4. Provide references

P12 L3-12: This is all described before! Provide references and try to discuss more about pre-event conditions

P12 L 8f: This finding is not new. Give references.

P12 L 13- P13 L13: Missing point: Conductivity in peatlands typically changes/decreases with depth! What depths are your piezometer? Give references! This is not a new topic! An important keyword here would be 'transmissivity feedback' or a similar effect.

P13 L 8 ff: This is nothing else than hydraulic conductivities

P13 L 18: introduce the term "hot moments"; what are the processes inducing hot moments?

P13 L 20: hydraulic conductivity? Acrotelm/catotelm in bog vs fen?

Fig. 1: Additionally, as your catchment is in a mountain area, it might be helpful to have some contour lines.

Fig. 2: showing an exemplary DOC event and way of examination is helpful. I would also like to see corresponding discharge values. In general an evaluation of DOC versus discharge (DOC/Q plot) might improve understanding of flow paths and DOC origin

Fig. 3: maybe add a line in the Bernadouze data set where the Ech dataset starts -> better comparability

Fig. 6: I am a bit lost what you want to show here. This is not how a peatland complex looks like and far too generalized trends that you cannot state like this.

Table 1: Header is not self-explanatory. Why did you choose peat water temperature at the beginning of DOC event and not weekly mean like at the other parameter?

Table 2: Check time period given for Ech. It seems to be incorrect (2015-2017?).

---

## Author Comment (AC2) · 15 Jan 2020

The manuscript of Thomas Rosset et al. aims at disentangling drivers of DOC dynamics in different peatland catchments. It utilizes a spectrofluorometric probe to monitoring frequency DOC concentrations. Additionally, parameters like temperature, precipitation, stream and peat water level were recorded to explain DOC concentration variability by means of statistic modelling. The generated dataset is quite extensive and worth to be published in Biogeosciences. As I was really pleased by the title and abstract I must say that the main part of the manuscript is rather descriptive and lacks a clear message or novelty. A major drawback of this manuscript is the lack of discussion of major processes and drivers concerning DOC export from peatlands, like hydrological flow paths, biogeochemistry and hydrologic preconditions. In its current state, the study is too much focused on the statistical results of their modelling, but no or little mechanistic explanation of the modelling results. What exactly happens during a rainfall/discharge/DOC concentration event in the different systems and concerning the hydrologic conditions? In general, I think that this paper needs a stronger discussion on hydrologic flow paths in peatland systems. Flow paths are not discussed until the very end. I miss a description of e.g. a simple acrotelm/catotelm distinction, which provide different hydraulic conductivities and thus lead to a distinct hydrograph. I miss a discussion of pre-event conditions, or of hydraulic conductivities in general, contributing to different flow paths. The interpretation of the piezometer data is difficult, as there is no information provided about depth of installation, hydraulic conductivities and thus how to interpret recession times. I miss mentioning importance of pH (DOC solubility!), do you have data about this? It is mentioned that the fen site is on limestone, while bog systems are generally acidic systems. As the authors are throwing in the term "bio-geochemical hotspot" in the end: I would be pleased to hear more about this earlier on in the manuscript. The introduction on second paragraph is rather superficial. Furthermore, I am very concerned by the representativeness of the bog site especially when it is compared to a fen as exemplary system (Scientific objective no 3, P3 line3). There are several factors differing between the sites, besides just fen/bog: climate (e.g. 4 months snow covered – no snow hardly sub-zero temperatures), anthropogenic influence (burned – unburned). Additionally, mentioning agro-pastoral practices: does this mean the bog is used for grazing? Could these systems thus be considered as representative? Besides this, from the location maps I draw conclusions that apparently the monitoring spot also receives water which is not originating from the peatlands itself. Is there any data about it? Do you have any idea about the whole catchment and how much water contributes to the discharge that is not from the peatlands? This is one of my major concerns, as I feel like the authors completely disregard this. If the concentration pattern are driven by discharge from other areas, the discussion of

concentration pattern and water levels at the monitoring spot and within the peatland would be difficult. Did you calculate also total DOC export fluxes next to the concentrations? This would highlight, how important this carbon output is to the system, as the concentrations seem to be quite low. Besides from this: Did you also compare the Bernadouze and Ech sites over a time period, where data from both sites are available? This would help to access, if there is any bias by having two different time periods here. Weather conditions can be very different each year having a very dry or wet year or season, re-spectively. I am no native speaker myself, so it is not easy for me to criticize language issues. But even though in your acknowledgement you state a language assistance,I am sure that there are some unusual or wrong wordings in the text (e.g. confusingus age of "contrasted" "strong concentrations"). So, from my point of view this needs further editing. Furthermore, your expression is imprecise at some points. Please edit (e.g. header of Table 1; P5 Lines 2-5: it is not clear what you mean; or speaking of "stream level increase") Please, check your figures for clear distinction and readability when not color printed (e.g. Fig 1 stream/peatland boundary, Fig. 5 Ech/Ber). In summary I see a very valuable dataset of potentially high interest. However, in its current state the discussion seems too superficial and the study remains rather descriptive. Therefore, I recommend major revisions before this study can be accepted for publication.

We thank the reviewer for constructive comments on our manuscript. Following the general comments, we have worked on improving the manuscript. First, we have improved the site description, including previous work (P4 L. 6). Then, we have included more details on the piezometers in the manuscript and added a table including installation depths, MRC, pH and DOC values for each study plot. Finally, we have improved the discussion section. We have tried to better emphasize the novelty and the contribution of our study (Section 5.1). We have included more discussion on hydrological processes that can be inferred from our study (Section 5.3 P11 L19 to P12 L9 ). We hope these improvements have clarified the manuscript. The answers to the specific comments can be found below.

P1 L 10/P2 L2: A very common number I know is 30% of global carbon stock. Please check more references for the number you give. Organic carbon stock in peatlands is a number constantly updated, at the submission time 20% was a figure obtained by crossing figures from Lefield and Menichetti, 2018 and Schalermann et al., 2014. According to the recent article from Nichols and Peteet, 2019, this figure is updated to 30%. We want to highlight the uncertainty of this figure writing ∼20-30%

P2 L 12-15: too simplistic statements. Drivers of DOC concentration are not dependent on latitude, but mostly on the system studied and climate The sentence has been removed.

P 2 L 32: "seasonal climatic conditions are contrasted" what does this mean? We meant that different abiotic parameters (temperature, precipitation, hydrology) evolves along both seasonal and event (snowmelt, rainstorms) scales. The sentence has been removed. Some details have been added in the text (P2 L. 33).

P3 L 19: What "logging activities" do you mean? Selective logging (1 tree over three was cut) was carried out during the autumn 2016 in the lowest forested area surrounding the peatland, producing no significant hydrological and biogeochemical offsets at the outlet of the peatland. These details have been added in the text P3 L20

P4 L 21: In which depths are the piezometers connected to the peat body? This is very important if you talk about recession times and peat water temperature. Maybe also interesting: What diameter have these wells? Piezometers wells are 50 mm diameter PVC tubes connected to the peat body at an average depth of 1.2 ±0.3 m in Bernadouze, except for two piezometers in the center of the peatland which are connected at 2.2 m depth, and at an average depth of 2.4 ± 0.1 m in Ech. PVC tubes are slotted from the bottom to 10 cm below the soil surface. These details have been added in a table as Appendix B and in the text (P4 L. 29)

Or did you attempt to determine hydraulic conductivities by a slug test? + comment P13 L 8 ff: This is nothing else than hydraulic conductivities + comment P13 L 20: hydraulic

conductivity? Acrotelm/catotelm in bog vs fen? We agree that MRC are proxies of the hydraulic conductivities. To characterize the hydrodynamic properties of the peat, MRC were preferred to hydraulic conductivity estimation from slug tests because they can be performed directly with the water table dataset. Thus, the proposed model could be more easily tested on other peatlands. Details have been added to the manuscript P6 L4 to explain our choice in the methodology and some sentences have been modified in the discussion to P13 L26/ P14 L15 to refer to hydraulic conductivity.

P5 L 2 "Flood sampling" is a weird expression. What about event sampling? Expression has been changed. P5 L9

P5 L 12: when did "the turbidity events" occur? I assume this is mostly the case during high discharge. How much of your DOC event data is affected by this? High turbidity events occurs occasionally at the beginning of high discharge events ( Rosset el al.,2019). The high turbidity period (> 20 FNU) are sporadic representing only 0,2% of the fDOM time series. Turbidity peaks do not affect the DOC event data, since DOC peaks occur after turbidity events. These details have been added in the manuscript (P5 L. 21).

P5 L 11: Reference not correctly inserted in text, happens occasionally in the manuscript The references have been modified (P5 L20 /P7 L8 / P7 L14)

P6 L 20: which variables were selected? Next sentences are unclear. The text has been rephrased for clarity (P7 L3).

P7 L 31 "...1.36 and 0.35. in..." unit missing The meter unit (m) has been added P8 L9

P7 L 19: why? You did not introduce site heterogeneity before (Introduction). I would also like to have an introduction in hydrologic connectivity, so are the different spots relevant for the discharge? What about the rest of the catchment? Site heterogeneity is discussed in the section 3.1 P4 L28. The first step of our investigation was based on an average peat water table (section 3.3), in order to explore the link with DOC

dynamics on the longest possible record. After the selection of the main explanatory variables, we investigated how different model performances between the bog and the fen with could be explained using the different recession times observed in the piezometer wells.

P8 L 4: Discuss different pre-conditions, changes in hydrologic conductivity with depth. This sentence aims at giving a general description of the water table depth at both site. This study focuses on DOC concentration peaks. Pre-conditions of each events were taken into account by defining explanatory variables integrating these pre-conditions as mentioned in Table 1 (water table level at the beginning of the DOC event, air and water temperature integrated on the 7 days prior the event, precipitation integrated on the 24 hours before the event, and time between peaks). Then, these pre-conditions are discussed in case they appeared as significant variable in the models.

P8 L 9: give conc. maxima. I am very surprised by the low mean values. When you have 2 and 7 mg/L mean DOC concentrations I am about to doubt the significance for the carbon balance. This only gives me the idea that you have a lot of water not originating from the peatlands itself. Calculating fluxes might be helpful here. You might check literature and compare. Our study sites are mountain peatlands, located in calcareous watersheds. The measured DOC concentration at the outlet are, as noticed by the reviewer, in the lower end of what can be expected in peatland sites. At the Bernadouze site, we performed an extensive study, including a high frequency survey at the inlet and outlet of the peatland. From this study (Rosset et al., 2019), we could conclude that the peatlands area contributed to 60 to 80 % of the fluvial carbon export. The specific fluxes estimated for the two sites range from 16.7$\pm$0.4 to 31.9$\pm$1.4 g.m$^{-2}$.yr-1 for bernadouze and 18.8$\pm$4.2 to 22$\pm$6.7 g.m$^{-2}$.yr -1 for Ech which is the high range of specific fluxes published for temperate peatlands and will be significant when establishing the carbon balance.

P8 L 31/32: Is this much? More than half cannot be explained by your model in Ech, other factors seem very important as well. Which could this be? The different model

performance between the two sites are discussed in terms of hydraulic conductivity ( MRC ) in the section 5.5 of the manuscript.

P 9 L 1: Peat water temperature dependent on depth? Peat water temperature was monitored in the piezometers and is representative of a mix of the whole water column.

P 9 L 9: speaking from "water table increase" and "water level increase" is confusing to me. Give more precise names, maybe include "stream" or "peat" for clear distinction The term stream and peat have been added when needed in the manuscript to make a clear distinction between water table and stream (section 4.4 and 5.5)

P9 L 19: This is a very simple statement and would be very odd, if this is not the case. This section describes the results, including simple statements like this one. However, the sentence has been slightly modified to shorten the description part (P9 L32).

P9 L 27: what is a "strong concentration"? The text has been modified to "higher concentration". P10 L 7

P9 L 29: Please reword The title of the section has been modified to 'Long term high frequency in situ monitoring" P10 L10

P9 L 30: Is this important? What is the novel statement/finding of your study? So far I mainly see confirmation of former results We do believe the high frequency survey is important for our results. First, without high frequency monitoring, the DOC peaks (< 32 hours) would never have been identified. This is a specificity of our mountain peatlands and such high numbers of events have never been reported. Then, the coupled analysis of DOC concentration and controlling parameter has allowed, to relate peat water table variation to DOC concentration at a very fine temporal scale. This, we are confident of, is a contribution to the literature since most study relate DOC and peat water table dynamic at the seasonal scale. The paragraph has been reworked to replace our study in the emerging literature on high frequency nutrient monitoring (P10 L. 11 to 21).

[Figure]

P 10 L 2 ff: Biogeochemistry? This is new and not mentioned before. Please introduce, the biogeochemical background or processes could be much more discussed in this paper As stated just above, the whole paragraph has been reworded.

P 10 L 5: expression "Thanks to" sounds uncommon, please reword. The sentence has been rephrased to 'This was possible with ...". P10 L19

P. 10 L 7 following: Chapter 5.2 is badly structured. If evapotranspiration is unlikely don't put it there so prominent. The discussion of this part is interrupted and starts again further below -> confusing This section has been restructured to start with the discussion with the seasonal control on DOC concentration and later discuss other hypotheses for our site. Moreover a new section 5.4 has been created to discuss about snow influence and to enhance the clarity of our manuscript.

P11 L 1: Peat pore water DOC: You did not mention those in the result section. A short description/discussion about those concentrations would be nice. The peat porewater sampling and characterization has been added in the method section (P5 L3). The average DOC and pH for each piezometer have been detailed in the table A2 included in the appendix section. The discussion on peat porewater DOC concentration has been moved to section 5.5 P12 L3.

P11 L 3 f: What differences? Explanatory variables? Leachable Pools is a good keyword, but please discuss this on your data and not switch directly to the next topic. The discussion on difference on porewater DOC concentrations between the two sites, as stated above, has been rephrased and move to the section 5.5.

P11 L 8. This is indicative for the discussion. There is just an isolated statement that vegetation type plays a role for DOC production. How is this related to your findings? Concerning the two last comments, this related discussion part has been moved to section 5.5 P12 L7 since we think that parameters such as recession time, pH and main vegetation cover are interdependent in peatlands, all being related to hydrology. In this manuscript, we chose to emphasize the relationship between DOC concentration
and the recession time since we assume that hydrology is the principal driver of the peatland biogeochemistry.

P 11 L 21: This is not true! Please try some more literature search This formulation was unfortunate and was modified in the manuscript P11 L14. We wanted to emphasize the scarcity of studies coupling high frequency data of DOC concentrations in the stream and peat water table depth.

P 11 L 22: Discuss flow paths. Try e.g. DOC concentration vs Q plots. You throw in "non-linear flow DOC concentration relationship", try to discuss this.

The DOC concentration vs Q plots are included here for the reviewer's reference. The DOC vs Q plots are included here for the reviewer's reference. No systematic relationship is observed between DOC concentration and discharge values.

P 11 L 30: just single observations, discuss mechanisms This section 5. 3 of the discussion has been expanded to describe the mechanisms involved during flood events. See P11 L30 to P12 L10.

P 12 L 3-4. Provide references + P12 L3-12: This is all described before! Provide references and try to discuss more about pre-event conditions + P12 L 8f: This finding is not new. Give references. Answer to the three comments above. This section has been restructured. The revised text includes references on the link between the volume of aerated peat and DOC production on one hand and DOC transfer in the other hand P12 L20-21.

P12 L 13- P13 L13: Missing point: Conductivity in peatlands typically changes/decreases with depth! What depths are your piezometer? Give references! This is not a new topic! An important keyword here would be 'transmissivity feedback' or a similar effect. To characterize the hydrodynamic properties of the peat, MRC where preferred to hydraulic conductivity estimation from slug tests because they can be performed directly with the water table dataset. Thus, the proposed model could be

more easily tested on other peatlands. Information regarding piezometer installation depths have been included in the revised manuscript, in the Appendix section (Table A2).

P13 L 18: introduce the term "hot moments"; what are the processes inducing hot moments The term "hot moment" has been introduced in the first section of the discussion (section 5.1 P10 L17) and refers to (McClain et al., 2003).

Fig. 1: Additionally, as your catchment is in a mountain area, it might be helpful to have some contour lines. Altitudes of the peatlands were added in the figure caption. We prefer not to add the contour lines, since the figure are focused on the peatland areas, and not the whole catchment.

Fig. 2: showing an exemplary DOC event and way of examination is helpful. I would also like to see corresponding discharge values. In general an evaluation of DOC versus discharge (DOC/Q plot) might improve understanding of flow paths and DOC origin The DOC vs Q plots have been included above. However, we prefer not to include them in the manuscript.

Fig. 3: maybe add a line in the Bernadouze data set where the Ech dataset starts ->better comparability We do not want to compare two similar period, since climatic conditions are not the same from one site to the other. Instead we chose to discuss the statistical models which are built in order to be independent from climatic variabilities.

Fig. 6: I am a bit lost what you want to show here. This is not how a peatland complex looks like and far too generalized trends that you cannot state like this. This figure is a conceptual schema to describe the two characteristic type of peat units (bog/fen) which may contribute to the DOC transfer at the outlet of a peatland complex. This was intentionally generalized to present the extremum of contribution in term of peat units, one with a low hydraulic conductivity (long recession time) and the second with a high hydraulic conductivity (short recession time).

Table 1: Header is not self-explanatory. Why did you choose peat water temperature at the beginning of DOC event and not weekly mean like at the other parameter? Peat water temperature does not vary as the air, or stream temperature do at the event scale. We chose the beginning of the DOC event since peat water temperature is an integrative variable, evolving mainly at the seasonal scale.

Table 2: Check time period given for Ech. It seems to be incorrect (2015-2017?) It was a mistake. As you mentioned, the monitored time period in Ech was 22nd May 2017 to 13th February 2019

Cited references: Leifeld, J., & Menichetti, L. (2018). The underappreciated potential of peatlands in global climate change mitigation strategies. Nature communications, 9(1), 1071. Nichols, J. E., & Peteet, D. M. (2019). Rapid expansion of northern peatlands and doubled estimate of carbon storage. Nature Geoscience, 12(11), 917-921. Rosset, T., Binet, S., Antoine, J. M., Lerigoleur, E., Rigal, F., & Gandois, L. Drivers of seasonal and event scale DOC dynamics at the outlet of mountainous peatlands revealed by high frequency monitoring. Scharlemann, J. P., Tanner, E. V., Hiederer, R. and Kapos, V.: Global soil carbon: understanding and managing the largest terrestrial carbon pool, Carbon Manag., 5(1), 81–91, doi:10.4155/cmt.13.77, 2014.

Please also note the supplement to this comment:
https://www.biogeosciences-discuss.net/bg-2019-372/bg-2019-372-AC2-supplement.pdf

[Figure]

[Figure]

**Fig. 1.** DOC concentration vs Q plots

---

## Author Response (AR1)

**Author's response to reviewer 1**

**Legend:**

- Reviewer comments in blue
- Author's responses in black
* * *
General comments: Rosset et al. reported high-resolution sensor data to investigate the mechanisms driving DOC concentration at the outlet of a bog and a fen in the French Pyrenees. The data and results are interesting. However, the paper can be improved further by explaining how complete are the sensor data, and providing discussion on how water temperature is related with the input and output of organic carbon in the bog and the fen. Specific comments are below, which the authors may consider when revising the manuscript.

We thank the reviewer for this overall positive evaluation of our manuscript. Following the reviewer's suggestions, we have improved the manuscript to include details on the data on which the analysis is based. The effect of temperature on DOC inputs and outputs have been clarified in the discussion section. The answers to the specific comments can be found below.

p. 4, line 10-24: What is the percent of data for which gap-filling models were used? Also, has there been any period of power outage? The merit of this paper is on the high-resolution 'sensor' data. Thus, the information is needed on the number (or the percentage) of data points that has been actually collected.

The gap filling represents between 5 and 80 % of data. Details have been included for each parameter in the manuscript (Section 3.1 P4 L15 to 26).

p. 5, line 8-: How accurate was the analysis? What was the recovery of the reference material?

The quantification limit was 1 mg. L-1. Above this value, the analytical uncertainty was evaluated to $\pm 0.1$ mg.L$^{-1}$. Reference material included ION-915 ([DOC]= $1,37 \pm 0,41$ mg C L$^{-1}$) and ION 96.4 ([DOC]= $4,64 \pm 0,70$ mg C L$^{-1}$) (Environment and Climate Change Canada, Canada). This was detailed P5 L 16 to 19

p. 5, line 12-: If the data with >20 FNU were ignored, what is the percentage of those "ignored" data points compared to the total?

Also, considering that [DOC] can be high with high flow, those data points are potentially important in interpreting the results. If included, could they change the conclusions?

The removed data related to high turbidity represent only 0,2% of the fDOM time series. In addition, the turbidity peaks occur before the fDOM peaks. Their removal from the timeserie does not affect our analysis. Some details have been added in the manuscript (P5 L23 to 26).

I think the graphs showing the relationship between the [DOC] and fDOM would be helpful. Can you add the graph as a supporting information?

Graphs showing the linear relationship between [DOC] and fDOM were added in Appendix A1

p. 5, line 16: number of observations 174 vs. 27. Why are these so different?

The different observation numbers are related to the different observation periods in the two sites. The survey started in 2015 in Bernadouze and in 2017 in Ech. Moreover, the number of flood event sampling (~20 samples each in average) differ between the two sites. Seven events could be sampled in Bernadouze, when only one was sampled in Ech. The number of observations at Ech was actually 28, it has been modified in the manuscript P5 L30

p. 5, line 27: what is the K in the equation 1? Please explain the terms in the equation.

K in the equation 1 was a constant. We replaced K by B in this second version of the manuscript to avoid confusion with K, commonly used in hydrology to described the hydraulic conductivity. Details have been added in the text to mention that B is constant P6 L14

p. 6, line 11: Have you used "DOC_max" for the analysis? If not, why didn't you include it for the analysis?

DOC_max was used in the analysis to calculate DOC_increase (DOC_increase= (DOC_max – DOC_initial)).

p. 7, line 3-: So, did log- or square root-transformation satisfy the assumption? Was non-parametric analysis unnecessary?

Prior to the analyses we checked the distribution of each variable using histograms and found substantial deviations from normality for some of them (mainly right-skewed distributions). Therefore, we transformed these variables using log-or square-root to approach normality (see Table 1 in the main text) considering that in linear modelling the point is not the reach strict normality of the data but to approximate normality in order to obtain satisfying distribution of the residuals i.e. Normality and Homoscedasticity of the error distribution (Venables & Ripley, 2002. Zuur et al. 2009). We then surveyed each best model using diagnostic plots in order to detect deviations from normality and homoscedasticity in the residuals and to identify outliers. No specific deviations and outlier were detected (See figures below) and we are therefore confident that our modelling approach and associated results are robust.

The use of non-parametric tests is always an option when normality assumption is grossly violated and when data-transformation cannot overcome deviation to normal distribution. However, non-parametric tests do not cope well with complex dataset and complex modelling. For instance, there is no non-parametric form of any regression. Regression means you are assuming that a particular parameterized model generated your data, and you try to find the parameters. Non-parametric tests are test that make no assumptions about the model that generated your data. Those two approaches are therefore incompatible.

In our study, we clearly favored parametric approaches in order propose hypothetical models explaining our two targeted variables i.e. DOC_increase and DOC_initial.

[Figure]

Figure review 1 Diagnostic plots of the DOC concentration increase in Bernadouze

[Figure]

Figure review 2 Diagnostic plots of the DOC concentration increase in Ech

[Figure]

Figure review 3 Diagnostic plots of the DOC concentration initial in Bernadouze

[Figure]

Figure review 4 Diagnostic plots of the DOC concentration initial in Ech

Fig. 1: Is the boundary of the watershed for the 'outlets' correct? Watershed boundary can be delineated for any point of a stream using DEM data. The watershed area for the red circles should be larger than the boundary of the fen or the bog (orange lines in Fig. 1). I wonder the DOC dynamics at the outlets could be significantly influenced by non-wetland areas considering that the stream lines are extended beyond the orange lines.

At both sites watershed boundaries have been delineated using DEM models, however only the peatland areas (3% of the watershed area in Bernadouze and 6% in Ech) were delineated on the figure 1 (orange lines). Peatlands are the main contributors of DOC at the outlets as reported in Rosset et al., 2019. This was explicitly written in the manuscript P4 L 7

Fig. 3: Is the purpose of this research on comparison between the fen and the bog? If so, which period should be used? The same overlapped period (May, 2017 to Jan 2019)? Or any period with available data? If you have chosen the second option (any period with available data) to maximize analysis power, why did you omit the period of Jan. 2015 to Sep. 2015 (Rosset et al., 2019, JGR-Bioigeosciences)?

The purpose of this research is not a direct comparison between a fen and a bog site. The purpose of this manuscript is to identify the drivers of the DOC concentration variability at peatland sites in general, so the period used for the analysis do not need to overlap. Moreover, the period between January and September 2015 was omitted in Bernadouze because almost 60% of the water table level sensors shot down during this period, preventing a good characterization of the mean water table level in the fen and consistent analysis.

Fig. 4: Interesting graphs. (a) When log (DOC initial) is∼2.0, the DOC initial should be∼100 mg/L. But, the maximum [DOC] in the Fig. 3d is∼30 mg/L. Why are these this so different? (b) What are the meanings of the y-intercept? When water table increase is 0, the log (DOC increase) is about -1 (fen) and +1 (bog). Then, DOC increase should be 0.1 mg/L (fen) and 10 mg/L (bog) even without the water table increase.

This is a mistake in the notation. The Logarithm (log) in this figure refers to natural logarithm, or neperian logarithm (ln (e) =1) and not as logarithm used with a base 10 (log 10 (10)=1. This has been corrected in an updated version of the figure.

What kind of mechanism is working? Concentration of DOC shows the dynamic balance between the input and output of organic carbon. How water temperature is related with the input and output of organic carbon in the bog and the fen?

We agree that concentration of DOC shows the dynamic balance between the input and output of organic carbon; However in these mountainous peatlands we observed that DOC concentrations are really lower at the inlet than at the outlet, as mentioned by Rosset et al., (2019). Thus, the mechanisms which control DOC concentration at the outlet occurs mainly within these peatlands and we did not consider that input of organic carbon from the inlet was a valuable variable to investigate, as input water temperature. However, the role of water temperature was investigated both within the peatland in the piezometer well and at the outlet in the stream. We highlight significant influence of peat temperature on seasonal variation of DOC concentration. This is discussed in detail at section 5.2.

Fig. 5: The graphs include many information and are hard to digest. I recommend to leave essential information only and provide the rest as a supporting material. Or figure caption can include in-detail explanation on the symbols

The figure caption has been modified for clarity. In addition, the legend in the figure has been enlarged.

**Cited references:**

Rosset, T., Gandois, L., Le Roux, G., Teisserenc, R., Durantez Jimenez, P., Camboulive, T., & Binet, S. (2019). Peatland contribution to stream organic carbon exports from a montane watershed. Journal of Geophysical Research: Biogeosciences.

Venables and Ripley. 2002. Modern Applied Statistics with S. Springer, New York. 4th edition.

Zuur, A., Ieno, E. N., Walker, N., Saveliev, A. A., & Smith, G. M. (2009). Mixed effects models and extensions in ecology with R. Springer Science

**Author's response to reviewer 2**

**Legend:**

- Reviewer comments in green
- Author's responses in black
* * *
The manuscript of Thomas Rosset et al. aims at disentangling drivers of DOC dynam-ics in different peatland catchments. It utilizes a spectrofluorometric probe to monitoring frequency DOC concentrations. Additionally, parameters like temperature, precipitation, stream and peat water level were recorded to explain DOC concentration variability by means of statistic modelling. The generated dataset is quite extensive and worth to be published in Biogeosciences. As I was really pleased by the title and abstract I must say that the main part of the manuscript is rather descriptive and lacks a clear message or novelty. A major drawback of this manuscript is the lack of discussion of major processes and drivers concerning DOC export from peatlands, like hydrological flow paths, biogeochemistry and hydrologic preconditions. In its current state, the study is too much focused on the statistical results of their modelling, but no or little mechanistic explanation of the modelling results. What exactly happens during a rainfall/discharge/DOC concentration event in the different systems and concerning the hydrologic conditions? In general, I think that this paper needs a stronger discussion on hydrologic flow paths in peatland systems. Flow paths are not discussed until the very end. I miss a description of e.g. a simple acrotelm/catotelm distinction, which provide different hydraulic conductivities and thus lead to a distinct hydrograph. I miss a discussion of pre-event conditions, or of hydraulic conductivities in general, contributing to different flow paths. The interpretation of the piezometer data is difficult, as there is no information provided about depth of installation, hydraulic conductivities and thus how to interpret recession times. I miss mentioning importance of pH (DOC solubility!), do you have data about this? It is mentioned that the fen site is on limestone, while bog systems are generally acidic systems. As the authors are throwing in the term "bio-geochemical hotspot" in the end: I would be pleased to hear more about this earlier on in the manuscript. The introduction on second paragraph is rather superficial. Furthermore, I am very concerned by the representativeness of the bog site especially when it is compared to a fen as exemplary system (Scientific objective no 3, P3 line3). There are several factors differing between the sites, besides just fen/bog: climate (e.g. 4 months snow covered – no snow hardly sub-zero temperatures), anthropogenic influence (burned – unburned). Additionally, mentioning agro-pastoral practices: does this mean the bog is used for grazing? Could these systems thus be considered as representative? Besides this, from the location maps I draw conclusions that apparently the monitoring spot also receives water which is not originating from the peatlands it-self. Is there any data about it? Do you have any idea about the whole catchment and how much water contributes to the discharge that is not from the peatlands? This is one of my major concerns, as I feel like the authors completely disregard this. If the concentration pattern are driven by discharge from other areas, the discussion of concentration pattern and water levels at the monitoring spot and within the peatland would be difficult. Did you calculate also total DOC export fluxes next to the concentrations? This would highlight, how important this carbon output is to the system, as the concentrations seem to be quite low. Besides from this: Did you also compare the Bernadouze and Ech sites over a time period, where data from both sites are available? This would help to access, if there is any bias by having two different time periods here. Weather conditions can be very different each year having a very dry or wet year or season, re-spectively. I am no native speaker myself, so it is not easy for me to criticize language issues. But even though in your acknowledgement you state a language assistance,I am sure that there are some unusual or wrong wordings in the text (e.g. confusingus age of "contrasted" "strong concentrations"). So, from my point of view this needs further editing. Furthermore, your expression is imprecise at some points. Please edit (e.g. header of Table 1; P5 Lines 2-5: it is not clear what you mean; or speaking of "stream level increase") Please, check your figures for clear distinction and readability when not color printed (e.g. Fig 1 stream/peatland boundary, Fig. 5 Ech/Ber). In summary I see a very valuable dataset of potentially high interest. However, in its current state the discussion seems too superficial and the study remains rather descriptive. Therefore, I recommend major revisions before this study can be accepted for publication.

We thank the reviewer for constructive comments on our manuscript. Following the general comments, we have worked on improving the manuscript. First, we have improved the site description, including previous work (P4 L. 7). Then, we have included more details on the piezometers in the manuscript and added a table including installation depths, MRC, pH and DOC values for each study plot. Finally, we have improved the discussion section. We have tried to better emphasize the novelty and the contribution of our study (Section 5.1). We have included more discussion on hydrological processes that can be inferred from our study (Section 5.3 P11 L22 to P12 L20 ). We hope these improvements have clarified the manuscript. The answers to the specific comments can be found below.

P1 L 10/P2 L2: A very common number I know is 30% of global carbon stock. Please check more references for the number you give.

Organic carbon stock in peatlands is a number constantly updated, at the submission time 20% was a figure obtained by crossing figures from Lefield and Menichetti, 2018 and Schalermann et al., 2014. According to the recent article from Nichols and Peteet, 2019, this figure is updated to 30%. We want to highlight the uncertainty of this figure writing ~20-30%

P2 L 12-15: too simplistic statements. Drivers of DOC concentration are not dependent on latitude, but mostly on the system studied and climate

The sentence has been removed.

P 2 L 32: "seasonal climatic conditions are contrasted" what does this mean?

We meant that different abiotic parameters (temperature, precipitation, hydrology) evolves along both seasonal and event (snowmelt, rainstorms) scales. The sentence has been removed. Some details have been added in the text (P2 L. 34).

P3 L 19: What "logging activities" do you mean?

Selective logging (1 tree out of three was cut) was carried out during the autumn 2016 in the lowest forested area surrounding the peatland, producing no significant hydrological and biogeochemical offsets at the outlet of the peatland. These details have been added in the text P3 L21

P4 L 21: In which depths are the piezometers connected to the peat body? This is very important if you talk about recession times and peat water temperature. Maybe also interesting: What diameter have these wells?

The piezometer wells are 50 mm diameter PVC tubes slotted from the bottom to 10 cm below the soil surface. The average depth in Bernadouze is about 1.2 ±0.3 m, except for two piezometers in the center of the peatland which were drilled to 2.2 m depth. The average depth of the Ech piezometers is 2.4 ± 0.1 m. These details have been added in a table as Appendix B  and in the text (P4 L. 30)

Or did you attempt to determine hydraulic conductivities by a slug test?

+  comment P13 L 8 ff: This is nothing else than hydraulic conductivities

+  comment P13 L 20: hydraulic conductivity? Acrotelm/catotelm in bog vs fen?

We agree that MRC are proxies of the hydraulic conductivities. To characterize the hydrodynamic properties of the peat, MRC were preferred to hydraulic conductivity estimation from slug tests because they can be performed directly with the water table dataset. Thus, the proposed model could be more easily tested on other peatlands. Details have been added to the manuscript P6 L8 to explain our choice in the methodology and some sentences have been modified in the discussion to P13 L31/ P14 L2 to refer to hydraulic conductivity.

P5 L 2 "Flood sampling" is a weird expression. What about event sampling?

Expression has been changed. P5 L11

P5 L 12: when did "the turbidity events" occur? I assume this is mostly the case during high discharge. How much of your DOC event data is affected by this?

High turbidity events occurs occasionally at the beginning of high discharge events ( Rosset el al.,2019). The high turbidity period (> 20 FNU) are sporadic representing only 0.2% of the fDOM time series. Turbidity peaks do not affect the DOC event data, since DOC peaks occur after turbidity events. These details have been added in the manuscript (P5 L. 23).

P5 L 11: Reference not correctly inserted in text, happens occasionally in the manuscript

The references have been modified (P5 L22 /P7 L11 / P7 L17)

P6 L 20: which variables were selected? Next sentences are unclear.

The text has been rephrased for clarity (P7 L6).

The meter unit (m) has been added P8 L11

P7 L 19: why? You did not introduce site heterogeneity before (Introduction). I would also like to have an introduction in hydrologic connectivity, so are the different spots relevant for the discharge? What about the rest of the catchment?

Site heterogeneity is discussed in the section 3.1 P4 L29.

The first step of our investigation was based on an average peat water table (section 3.3), in order to explore the link with DOC dynamics on the longest possible record. After the selection of the main explanatory variables, we investigated how different model performances between the bog and the fen with could be explained using the different recession times observed in the piezometer wells.

P8 L 4: Discuss different pre-conditions, changes in hydrologic conductivity with depth.

This sentence aims at giving a general description of the water table depth at both site. This study focuses on DOC concentration peaks. Pre-conditions of each events were taken into account by defining explanatory variables integrating these pre-conditions as mentioned in Table 1 (water table level at the beginning of the DOC event, air and water temperature integrated on the 7 days prior the event, precipitation integrated on the 24 hours before the event, and time between peaks). Then, these pre-conditions are discussed in case they appeared as significant variable in the models.

P8 L 9: give conc. maxima. I am very surprised by the low mean values. When you have 2 and 7 mg/L mean DOC concentrations I am about to doubt the significance for the carbon balance. This only gives me the idea that you have a lot of water not originating from the peatlands itself. Calculating fluxes might be helpful here. You might check literature and compare.

P8 – L21 Our study sites are mountain peatlands, located in calcareous watersheds. The measured DOC concentration at the outlet are, as noticed by the reviewer, in the lower end of what can be expected in peatland sites. At the Bernadouze site, we performed an extensive study, including a high frequency survey at the inlet and outlet of the peatland. From this study (Rosset et al., 2019), we could conclude that the peatlands area contributed to 60 to 80 % of the fluvial carbon export. The specific fluxes estimated for the two sites range from $16.7\pm0.4$ to $31.9\pm1.4$ g.m$^{-2}$.yr$^{-1}$ for bernadouze and $18.8\pm4.2$ to

$22\pm6.7$ g.m$^{-2}$.yr$^{-1}$ for Ech which is the high range of specific fluxes published for temperate peatlands and will be significant when establishing the carbon balance.

P8 L 31/32: Is this much? More than half cannot be explained by your model in Ech, other factors seem very important as well. Which could this be?

The different model performance between the two sites are discussed in terms of hydraulic conductivity ( MRC ) in the section 5.5 of the manuscript.

P 9 L 1: Peat water temperature dependent on depth?

Peat water temperature was monitored in the piezometers and is representative of a mix of the whole water column.

P 9 L 9: speaking from "water table increase" and "water level increase" is confusing to me. Give more precise names, maybe include "stream" or "peat" for clear distinction

The term stream and peat have been added when needed in the manuscript to make a clear distinction between water table and stream (section 4.4 and 5.5)

P9 L 19: This is a very simple statement and would be very odd, if this is not the case.

This section describes the results, including simple statements like this one. However, the sentence has been slightly modified to shorten the description part (P10 L2).

P9 L 27: what is a "strong concentration"?

The text has been modified to "higher concentration". P10 L 10

P9 L 29: Please reword

The title of the section has been modified to 'Long term high frequency in situ monitoring" P10 L13

P9 L 30: Is this important? What is the novel statement/finding of your study? So far I mainly see confirmation of former results

We do believe the high frequency survey is important for our results. First, without high frequency monitoring, the DOC peaks (< 32 hours) would never have been identified. This is a specificity of our mountain peatlands and such high numbers of events have never been reported. Then, the coupled analysis of DOC concentration and controlling parameter has allowed, to relate peat water table variation to DOC concentration at a very fine temporal scale. This, we are confident of, is a contribution to the literature since most study relate DOC and peat water table dynamic at the seasonal scale.

The paragraph has been reworked to replace our study in the emerging literature on high frequency nutrient monitoring (P10 L. 14 to 24).

P 10 L 2 ff: Biogeochemistry? This is new and not mentioned before. Please introduce, the biogeochemical background or processes could be much more discussed in this paper

P 10 L 5: expression "Thanks to" sounds uncommon, please reword.

As stated just above, the whole paragraph has been reworded.

P. 10 L 7 following: Chapter 5.2 is badly structured. If evapotranspiration is unlikely don't put it there so prominent. The discussion of this part is interrupted and starts again further below -> confusing

This section has been restructured to start with the discussion with the seasonal control on DOC concentration and later discuss other hypotheses for our site. Moreover a new section 5.4 has been created to discuss about snow influence and to enhance the clarity of our manuscript.

P11 L 1: Peat pore water DOC: You did not mention those in the result section. A short description/discussion about those concentrations would be nice.

The peat porewater sampling and characterization has been added in the method section (P5 L5). The average DOC and pH for each piezometer have been detailed in the table A2 included in the appendix section. The discussion on peat porewater DOC concentration has been moved to section 5.5 P12 L14.

P11 L 3 f: What differences? Explanatory variables? Leachable Pools is a good keyword, but please discuss this on your data and not switch directly to the next topic.

The discussion on difference on porewater DOC concentrations between the two sites, as stated above, has been rephrased and move to the section 5.5.

P11 L 8. This is indicative for the discussion. There is just an isolated statement that vegetation type plays a role for DOC production. How is this related to your findings?

Concerning the two last comments, this related discussion part has been moved to section 5.5 P12 L14 since we think that parameters such as recession time, pH and main vegetation cover are interdependent in peatlands, all being related to hydrology. In this manuscript, we chose to emphasize the relationship between DOC concentration and the recession time since we assume that hydrology is the principal driver of the peatland biogeochemistry.

P 11 L 21: This is not true! Please try some more literature search

This formulation was unfortunate and was modified in the manuscript P11 L28. We wanted to emphasize the scarcity of studies coupling high frequency data of DOC concentrations in the stream and peat water table depth.

P 11 L 22: Discuss flow paths. Try e.g. DOC concentration vs Q plots. You throw in "non-linear flow DOC concentration relationship", try to discuss this.

[Figure]

[Figure]

The DOC vs Q plots are included here for the reviewer's reference. The DOC vs Q plots are included here for the reviewer's reference. No systematic relationship is observed between DOC concentration and discharge values.

P 11 L 30: just single observations, discuss mechanisms

This section 5. 3 of the discussion has been expanded to describe the mechanisms involved during flood events. See P11 L22 to P12 L22.

P 12 L 3-4. Provide references

+

P12 L3-12: This is all described before! Provide references and try to discuss more about pre-event conditions

+

P12 L 8f: This finding is not new. Give references.

Answer to the three comments above. This section has been restructured. The revised text includes references on the link between the volume of aerated peat and DOC production on one hand and DOC transfer in the other hand P12 L5.

P12 L 13- P13 L13: Missing point: Conductivity in peatlands typically changes/decreases with depth! What depths are your piezometer? Give references! This is not a new topic! An important keyword here would be 'transmissivity feedback' or a similar effect.

To characterize the hydrodynamic properties of the peat, MRC where preferred to hydraulic conductivity estimation from slug tests because they can be performed directly with the water table dataset. Thus, the proposed model could be more easily tested on other peatlands.

Information regarding piezometer installation depths have been included in the revised manuscript, in the Appendix section (Table A2).

P13 L 18: introduce the term "hot moments"; what are the processes inducing hot moments

The term "hot moment" has been introduced in the first section of the discussion (section 5.1 P10 L20) and refers to (McClain et al., 2003).

Fig. 1: Additionally, as your catchment is in a mountain area, it might be helpful to have some contour lines.

Altitudes of the peatlands were added in the figure caption. We prefer not to add the contour lines, since the figure are focused on the peatland areas, and not the whole catchment.

Fig. 2: showing an exemplary DOC event and way of examination is helpful. I would also like to see corresponding discharge values. In general an evaluation of DOC versus discharge (DOC/Q plot) might improve understanding of flow paths and DOC origin

The DOC vs Q plots have been included above. However, we prefer not to include them in the manuscript.

Fig. 3: maybe add a line in the Bernadouze data set where the Ech dataset starts ->better comparability We do not want to compare two similar period, since climatic conditions are not the same from one site to the other. Instead we chose to discuss the statistical models which are built in order to be independent from climatic variabilities.

Fig. 6: I am a bit lost what you want to show here. This is not how a peatland complex looks like and far too generalized trends that you cannot state like this.

This figure is a conceptual schema to describe the two characteristic type of peat units (bog/fen) which may contribute to the DOC transfer at the outlet of a peatland complex. This was intentionally generalized to present the extremum of contribution in term of peat units, one with a low hydraulic conductivity (long recession time) and the second with a high hydraulic conductivity (short recession time).

Table 1: Header is not self-explanatory. Why did you choose peat water temperature at the beginning of DOC event and not weekly mean like at the other parameter?

Peat water temperature does not vary as the air, or stream temperature do at the event scale. We chose the beginning of the DOC event since peat water temperature is an integrative variable, evolving mainly at the seasonal scale.

Table 2: Check time period given for Ech. It seems to be incorrect (2015-2017?)

It was a mistake. As you mentioned, the monitored time period in Ech was 22nd May 2017 to 13th February 2019

**Cited references:**

Leifeld, J., & Menichetti, L. (2018). The underappreciated potential of peatlands in global climate change mitigation strategies. *Nature communications*, *9*(1), 1071.

Nichols, J. E., & Peteet, D. M. (2019). Rapid expansion of northern peatlands and doubled estimate of carbon storage. *Nature Geoscience*, *12*(11), 917-921.

Rosset, T., Binet, S., Antoine, J. M., Lerigoleur, E., Rigal, F., & Gandois, L. Drivers of seasonal and event scale DOC dynamics at the outlet of mountainous peatlands revealed by high frequency monitoring.

Scharlemann, J. P., Tanner, E. V., Hiederer, R. and Kapos, V.: Global soil carbon: understanding and managing the largest terrestrial carbon pool, Carbon Manag., 5(1), 81–91, doi:10.4155/cmt.13.77, 2014.

**Author's response to associate editor**

**Legend:**

- Associate editor's comments in brown
- Author's responses in black

I agree with the second reviewer that you need to strengthen the discussion of controlling mechanisms for the hydrologic export of DOC. In particular, I wondered if your statement "stream water level only poorly contributed to explaining the variability of DOC increases during flood events 25 (Table 3 and Fig.5 a)" would be valid, given the significant p level and, as you mentioned, the "non-linear flow-DOC concentration relationship". The non-linear hysteresis relationship between discharge and DOC concentrations might lower r square values, but you cannot say that discharge relationships do not play a role in hydrologic DOC export. Please refer to papers on this hysteresis discharge relationship and check out how your data conform to or depart from the reported patterns. An upland case based on high-resolution sensor data: Jeong et al.: Differential storm responses of dissolved and particulate organic carbon in a mountainous headwater stream, investigated by high-frequency, in situ optical measurements, J. Geophys. Res.-Biogeosci., 117, G03013, doi:10.1029/2012JG001999, 2012.

We thank the associate editor for adding new comments on our manuscript. In the new version of the manuscript we highlighted that our study was a statistical investigation of a large dataset to infer main abiotic parameters controlling stream DOC concentration at the outlet of peatlands. This a first step to understand the whole mechanism driving the DOC exports from peatlands and a preliminary study before creating mechanistic models (as suggested by reviewer 2). However, discussion has been strengthened to give more details about the potential mechanisms driving the DOC export at the outlet of the peatlands, in particular explaining the DOC flush occurring in the acrotelm when water table increase during a flood event.

Besides, regarding to the specific comment about the contribution of stream water level to stream DOC concentration prediction, we would like to emphasize that the sentence refers to the $R^2$ contribution of the stream water level in the MLR model. This figure is obtained using the hierarchical variance partitioning (Chevan and Sutherland, 1991) (See section 3.6). Even if this parameter is significant according to the p-value, its contribution is statistically poor compared to other parameters such as water table increase ($R^2$ contribution in Table 3).

p11 l22 Few sentences dealing with non-linearity and hysteretic patterns between DOC concentration and discharge has been incorporated in the manuscript. The suggested publication has been mentioned as other references about hysteresis analysis which are useful to describe mechanisms of nutrient exports but as mentioned in the manuscript that are not reliable DOC concentration predictors.

Finally, as suggested by the reviewer 2, the manuscript has been reviewed to correct language mistakes and improve the written English.

References

[revised manuscript text omitted]

---

## Referee Report (RR1)

I am happy to see that the authors thoroughly responded to the comments. I acknowledge the effort and see that the manuscript improved. However, there are a few points that have not been answered and I would like to see some short statement addressing these points in the manuscript before publishing:

1. *"Furthermore, I am very concerned by the representativeness of the bog site especially when it is compared to a fen as exemplary system (Scientific objective no 3, P3 line 3). There are several factors differing between the sites, besides just fen/bog: climate (e.g. 4 month snow covered – no snow hardly sub-zero temperatures), anthropogenic influence (burned – unburned). Additionally, mentioning agro-pastoral practices: does this mean the bog is used for grazing? Could these systems thus be considered as representative? Besides this, from the location maps I draw conclusions that apparently the monitoring spot also receives water which is not originating from the peatlands itself. Is there any data about it? Do you have any idea about the whole catchment and how much water contributes to the discharge that is not from the peatlands? This is one of my major concerns, as I feel like the authors completely disregard this. If the concentration pattern are driven by discharge from other areas, the discussion of concentration pattern and water levels at the monitoring spot and within the peatland would be difficult."*

   You haven't responded at all to these points. I at least would like you to clarify the role of additional water sources in your concentration pattern. I am in line with you that most DOC is originating from the bog, but this does not mean that there is no other main water source with a different hydrograph dynamic/respond to rain events and dry periods. To evaluate concentrations is something different than fluxes. This would bias your concentration pattern. I read in the respond to reviewer #1 that the peatland area is just covering 3 or 6 % of the catchment area, respectively. I want to read this in the site description and an explanation that you still see only a peatland signal there (similar to statement P4 L7) or a limiting statement also in the discussion. I don't like that this is completely disregarded.

2. I am glad that your discussion about hydrologic pathways and non-linear responses largely improved. I actually miss the reference of Birkel et al. 2017 (that you cited in Rosset et al. 2019). As this publication also modelled peatland water level and stream DOC concentrations and it seems that their findings are in line with your study. The outcome of that study might be worth discussed in view of your model results.
   I looked at your DOC/Q plots with great interest and I just want to note that I am impressed by this data set and that I am convinced that there is so much more information in it, which would be also worth to be evaluated (hysteresis loops, DOC pool dilution/exhaustion effects, different responses by different preconditions) and would greatly improve understanding of DOC export from peatlands. Maybe you can consider this for a future (meta)study.

---

## Author Response (AR2)

**Author's response to the associate editor - #2**

**Legend:**

- Associate editor comments in red
- Author's responses in black
* * *
Dear Authors,

Thank you for thoroughly revising your manuscript. The two reviewers had provided a substantial number of detailed technical comments during the first round of review, so I had to ask them to examine whether your revision had adequately addressed all the reviewer comments and suggestions.

Both reviewers have provided positive evaluations and I agree to their opinion that you had put tremendous efforts to improve the manuscript. However, the reviewers raised a few points that remain to be clarified further before the final acceptance. Please clarify these points (provided in the attached reviewer reports) in your final revision.

I would like to ask you to make all the changes easily identifiable in a marked-up manuscript based on your point-by-point responses to the comments offered by the two reviewers. If possible, please specify the line numbers of the revised parts in your responses to the reviewer comments.

Sincerely,

Ji-Hyung Park
Associate Editor, Biogeosciences

Dear Associate Editor,

We thank you for giving us the opportunity to improve this manuscript a second time. In this third version of the manuscript we corrected and clarified the few points that the reviewers had raised during their second review. As asked, the line numbers of the revised parts are specified in the following responses to the reviewers.

Sincerely,

The authors:

Thomas Rosset, Stéphane Binet, Jean-Marc Antoine, Emilie Lerigoleur, François Rigal and Laure Gandois

**Author's response to reviewer 1 - #2**

**Legend:**

- Reviewer comments in blue
- Author's responses in black
* * *
The authors have now clarified most of the issues raised by me. The explanation and approach are plausible. However, the second reviewer raised many valuable points. Although I have dealt with sensor data, I am not an expert on peatland ecosystem per se. Thus, I would value the second reviewer's opinion on the revised discussion.

Minor corrections are still need to be made.

We thank the reviewer for this second evaluation of our manuscript. We corrected the manuscript according to his comments. Our answers to the specific questions can be found below.

p. 5, line 8: what do you mean by "the first millimeters?"

(p5, Line 6) This is a mistake. We filtered a volume of water then millimeters has been changed to milliliters.

p. 5, line 18: I guess that ION 915 and ION 96.4 are reference materials. So, what was the range of recovery for the ION 915 and ION 96.4? In other words, what was the range of measured [DOC] for ION 915 and ION 96.4? The concentration range of the samples is higher than concentration of ION 915 and ION 96.4. Did you dilute the samples? Also, you mentioned that analytical uncertainty was estimated at 0.1 mg/L. Is "(+/- 0.41 mg/L)" reported value? What do you mean by the two decimal places?

ION 915 and ION 96.4 are reference materials which are certified with an accuracy of two decimal places (e.g +/- 0.41 mg/L). This accuracy is higher than the one of our analyzer which is defined at 0.1 mg/L.

The certified values do not define the calibration interval. The DOC concentration measurements are performed as follows. Prior the analysis, the organic carbon analyzer is calibrated using different sodium hydrogen phthalate solution ranging from 0.2 to 10.0 mg/L. During the analysis, samples higher than 10 mg/L are automatically diluted by the analyzer to measure concentration within the calibration range. Independently from the calibration range, reference material (ION96.4, ION 915) are introduced in the batch of samples and analyzed to certify the accuracy of the calibration We do not mention this analytical description in the manuscript since we consider it is far from the scope of this study and commonly used in biogeochemical laboratories.

p. 8, line 20: "pH" is missing.

p 8, line 18 This omission was corrected, mentioning pH values between brackets as follows (pH=5.0±0.4).

Table A2: use of ","? Two decimal places in some numbers need to be cut to one decimal place.

All the decimal were corrected in the table A2. Decimal for piezometer depths were also corrected in the manuscript (p4 l30 and 31) in order to be adjusted with the depths mentioned in the table A2.

**Author's response to reviewer 2 - #2**

**Legend:**

- Reviewer comments in blue
- Author's responses in black
* * *
I am happy to see that the authors thoroughly responded to the comments. I acknowledge the effort and see that the manuscript improved. However, there are a few points that have not been answered and I would like to see some short statement addressing these points in the manuscript before publishing:

We want to thank the reviewer for the precious comments he addresses about the second version of this manuscript The revised parts are marked up and commented further down in this document.

1. "Furthermore, I am very concerned by the representativeness of the bog site especially when it is compared to a fen as exemplary system (Scientific objective no 3, P3 line 3). There are several factors differing between the sites, besides just fen/bog: climate (e.g. 4 months snow covered – no snow hardly sub-zero temperatures), anthropogenic influence (burned – unburned). Additionally, mentioning agro-pastoral practices: does this mean the bog is used for grazing? Could these systems thus be considered as representative? Besides this, from the location maps I draw conclusions that apparently the monitoring spot also receives water which is not originating from the peatlands itself. Is there any data about it? Do you have any idea about the whole catchment and how much water contributes to the discharge that is not from the peatlands? This is one of my major concerns, as I feel like the authors completely disregard this. If the concentration pattern are driven by discharge from other areas, the discussion of concentration pattern and water levels at the monitoring spot and within the peatland would be difficult."

(Abstract p1; l22 + p3; L3) We agree that, in the first version of this manuscript, the scientific objective n°3 (to compare DOC concentration at the outlet of a bog and a fen) was miswritten. The two sites are representative of the peatlands observed in the Pyrenees and they present contrasted hydrological functioning. However, the reviewer is right when he mentions the anthropogenic and climatic conditions are not identical between the two sites. Then, in the introduction, the third objective was rephrased to "two contrasted peatlands regarding their hydrological functioning".

You haven't responded at all to these points. I at least would like you to clarify the role of additional water sources in your concentration pattern. I am in line with you that most DOC is originating from the bog, but this does not mean that there is no other main water source with a different hydrograph dynamic/respond to rain events and dry periods. To evaluate concentrations is something different than fluxes. This would bias your concentration pattern. I read in the respond to reviewer #1 that the peatland area is just covering 3 or 6 % of the catchment area, respectively. I want to read this in the site description and an explanation that you still see only a peatland signal there (similar to statement P4 L7) or a limiting statement also in the discussion. I don't like that this is completely disregarded.

(p3; l16 and l 24) The percentage of peatland cover in the two watersheds has been added in the site description.

(p12; l22) In the first version of this manuscript dilution of the DOC exported from the peatland was not neglected, but maybe poorly explained. According to the hydrographs that we observed at the outlet of the peatlands, we consider that the catchments contribute uniformly in term of water to the discharge at the outlet. As asked by the reviewer, these dilution processes of DOC in the water flowing from the watersheds were highlighted in the discussion according this hypothesis.

2. I am glad that your discussion about hydrologic pathways and non-linear responses largely improved. I actually miss the reference of Birkel et al. 2017 (that you cited in Rosset et al. 2019). As this publication also modelled peatland water level and stream DOC concentrations and it seems that their findings are in line with your study. The outcome of that study might be worth discussed in view of your model results.

(p11; l17). We agree that the article of Birkel et al 2017 is of a great interest to understand DOC concentration at the outlet of peatlands, especially regarding the non-linear links between Q and DOC concentration. The reference was added to mention a non-linear Q-DOC model example in the manuscript + (p12; l10). Moreover, the model of Birkel et al. 2017 was mentioned in the discussion since it emphasizes the link between the DOC concentration and the water table dynamic in the upper soil horizon.

I looked at your DOC/Q plots with great interest and I just want to note that I am impressed by this data set and that I am convinced that there is so much more information in it, which would be also worth to be evaluated (hysteresis loops, DOC pool dilution/exhaustion effects, different responses by different preconditions) and would greatly improve understanding of DOC export from peatlands. Maybe you can consider this for a future (meta)study.

We confirm that high frequency monitoring generate really interesting data sets to improve the biogeochemical and hydrological understanding of peatland ecosystems. Monitoring systems are still running and certainly, different analytical methods could be applied on this data set. This has to be taken into account for future publications and meta-project about peatlands.

[revised manuscript text omitted]